# *Campylobacter jejuni* genotypes are associated with post-infection irritable bowel syndrome in humans

Stephanie Peters [1,7], Ben Pascoe [2,7], Zuowei Wu [3], Sion C. Bayliss[2], Ximin Zeng[1], Adam Edwinson[1], Sakteesh Veerabadhran-Gurunathan[1], Selina Jawahir[4], Jessica K. Calland[2], Evangelos Mourkas[2], Robin Patel[5], Terra Wiens[5], Marijke Decuir[5], David Boxrud[5], Kirk Smith[5], Craig T. Parker[6], Gianrico Farrugia [1], Qijing Zhang[3], Samuel K. Sheppard[2✉] & Madhusudan Grover [1✉]

*Campylobacter* enterocolitis may lead to post-infection irritable bowel syndrome (PI-IBS) and while some *C. jejuni* strains are more likely than others to cause human disease, genomic and virulence characteristics promoting PI-IBS development remain uncharacterized. We combined pangenome-wide association studies and phenotypic assays to compare *C. jejuni* isolates from patients who developed PI-IBS with those who did not. We show that variation in bacterial stress response (*Cj0145_phoX*), adhesion protein (*Cj0628_CapA*), and core biosynthetic pathway genes (biotin: *Cj0308_bioD*; purine: *Cj0514_purQ*; isoprenoid: *Cj0894c_ispH*) were associated with PI-IBS development. In vitro assays demonstrated greater adhesion, invasion, IL-8 and TNFα secretion on colonocytes with PI-IBS compared to PI-no-IBS strains. A risk-score for PI-IBS development was generated using 22 genomic markers, four of which were from *Cj1631c*, a putative heme oxidase gene linked to virulence. Our finding that specific *Campylobacter* genotypes confer greater in vitro virulence and increased risk of PI-IBS has potential to improve understanding of the complex host-pathogen interactions underlying this condition.

[1] Department of Gastroenterology and Hepatology, Mayo Clinic, Rochester, MN, USA. [2] The Milner Centre for Evolution, Department of Biology and Biochemistry, University of Bath, Claverton Down, Bath, UK. [3] Department of Veterinary Microbiology and Preventive Medicine, Iowa State University, Ames, IA, USA. [4] Minnesota Department of Health, St. Paul, MN, USA. [5] Division of Clinical Microbiology, Department of Laboratory Medicine and Pathology, Mayo Clinic, Rochester, MN, USA. [6] United States Department of Agriculture, Albany, CA, USA. [7]These authors contributed equally: Stephanie Peters, Ben Pascoe. ✉email: s.k.sheppard@bath.ac.uk; Grover.Madhusudan@mayo.edu

Irritable bowel syndrome (IBS) is a chronic, disabling gastro-intestinal disorder that affects up to 15% of people worldwide[1]. Patients suffer from chronic symptoms of abdominal pain and alteration in frequency and/or form of the stool. Acute infectious gastroenteritis is a major risk factor for development of this condition with the onset of chronic symptoms often emerging post-infection (PI-IBS)[2]. Most PI-IBS patients have abdominal pain with either diarrhea or alternating episodes of diarrhea and constipation[3]. *Campylobacter* species are among the most common causes of bacterial enterocolitis worldwide and estimates of PI-IBS rates associated with this enteric pathogen are greater than 20%[3,4]. It is known that the severity of *Campylobacter* enterocolitis is linked to the likelihood of developing PI-IBS. For example, bloody stools, hospitalization, and duration of acute illness are known to be associated with increased risk of PI-IBS development[3,5,6]. However, while *Campylobacter* species and strains vary in importance as gastrointestinal pathogens, the lineages associated with PI-IBS and underlying genetic drivers are poorly understood.

Over 90% of human *Campylobacter* infections are caused by *C. jejuni* or *C. coli*, mostly acquired through consumption of contaminated meat and poultry[7]. Acute intestinal injury from *Campylobacter* species as well as the host response play a role in the pathogenesis of PI-IBS. In vitro studies have shown that *C. jejuni* can invade epithelial cells[8,9], disrupt tight junctions[8] and affect intestinal fluid transport[10]. In severe cases, acute infection can lead to impairment of the colonic barrier and absorptive function[11]. It is clear that not all *C. jejuni* strains are equally important as human pathogens, but the extent to which this reflects different infection dynamics or variable pathogenicity among strains is not well known[12,13].

There is evidence that some strains are carried asymptomatically[14] while others are associated with sequelae such as reactive arthritis or Guillain-Barre syndrome[15]. Furthermore, in vitro assays have shown variation in the propensity of strains to invade epithelial cells and invoke pro-inflammatory cytokine (IL-8) response[16,17]. Key to explaining the differential pathogenicity among strains is gaining an understanding of the genetic basis of phenotypic variation, specifically the genes that underlie traits conferring pathogenic potential. Outer membrane protein, toxin production, and flagella biosynthesis genes have all been implicated as virulence determinants[17–21]. However, to date, no single gene, plasmid, phage or pathogenicity island has been causally linked to acute illness in humans.

In this study, using *Campylobacter* isolates from a well-characterized cohort of patients who developed *Campylobacter* PI-IBS and those who did not, we identify sequence types and lineages associated with development of PI-IBS. Applying a genome-wide association study (GWAS) approach linked to clinically relevant phenotypes tested in vitro, we identify genes and genetic elements associated with pathogenicity and prediction of PI-IBS development. This approach has the potential to improve understanding of the genetic determinants of virulence and allows the calculation of a risk score for individual genotypes that, with further validation, could be a basis for medical interventions aimed at preventing PI-IBS following *Campylobacter* infection.

## Results

**Campylobacter isolates associated with PI-IBS cases and controls.** As a part of statewide surveillance, the Minnesota Department of Health collects data on symptoms and exposures within 7–10 days upon notification of culture-positive *Campylobacter* cases and collects fecal samples. For this study, from 2011 through 2019, we prospectively sent surveys (Rome III and IBS symptom severity) to patients 6–9 months after *Campylobacter*

infection. Patients with IBS prior to the infection or history of inflammatory bowel disease, celiac disease, and microscopic colitis were excluded from the prospective survey. The Rome III criteria requires presence of abdominal pain or discomfort at least 2–3 times a month and alterations in bowel function (stool consistency or frequency) for ≥6 months to demonstrate chronicity of symptoms[3,22]. Ninety-four isolates were collected from the Minnesota Department of Health and sequenced, of which 88 were *C. jejuni* and 6 *C. coli* (Supplementary Data 1). Infection with 49 of the *C. jejuni* isolates led to PI-IBS, whereas 30 did not (defined as control or PI-no-IBS isolates). Nine *C. jejuni* isolates lacked complete follow up data. The mean (SD) IBS symptom severity score (0-500 scale) for PI-IBS was 182.6 (117.6) suggesting moderate IBS symptom severity (defined by a score range 175–300)[23]. Demographic and clinical details of PI-IBS patients and controls are provided in Table 1.

**Acute Campylobacter infection isolates are genetically diverse.** A maximum likelihood phylogeny constructed from a concatenated gene-by-gene core genome alignment revealed a highly structured population ($n = 94$; Fig. 1a, Supplementary Data 2). Isolates shared 1225 core genes (loci present in >95% of genomes; equivalent to 75.7% of the NCTC11168 reference genome). *C. jejuni* ($n = 88$) and *C. coli* ($n = 6$) isolates clustered into 59 common sequence types (ST), as defined by seven MLST genes[24] (Fig. 1a; Supplementary Data 3). Acute infection was predominantly caused by host generalist lineages, previously associated with more than one source reservoir ($n = 25$ for ST21 and ST45, most common clonal complex isolates); and poultry specialist ($n = 18$ for ST353, ST354 most common clonal complexes) (Fig. 1b)[25]. Fewer acute infection isolates were from ruminant-associated lineages (Supplementary Fig. 1A, B).

**PI-IBS emerges from multiple C. jejuni lineages.** Risk of PI-IBS following infection by isolates from different lineages varied (36 STs for PI-IBS and 21 STs for controls), with 18 different STs representing both PI-IBS and control *C. jejuni*. The ST922 clonal complex isolates were consistently associated with PI-IBS (3 of 3

**Table 1 Demographic and clinical characteristics of patients during acute *C. jejuni* gastroenteritis.**

| | PI-IBS ($n = 49$) | Control ($n = 30$) |
|---|---|---|
| Age, years | 39 ± 2 | 43 ± 2 |
| Sex, F/M | 34/15 | 21/9 |
| Abdominal cramps | 38/41 (93%) | 21/25 (84%) |
| Diarrhea | 44/45 (98%) | 30/30 (100%) |
| Average bowel movements/day[a] | 16 ± 2.3 | 16 ± 2.4 |
| Duration of diarrhea, days[b] | 8.7 ± 0.9 | 6.7 ± 0.7 |
| Blood in the stool | 18/45 (40%) | 9/30 (30%) |
| Nausea | 28/40 (70%) | 19/25 (76%) |
| Vomiting | 17/40 (42%) | 7/25 (28%) |
| Fever | 32/45 (71%) | 24/30 (80%) |
| Temperature (°F)[c] | 101.8 ± 0.3 | 102.1 ± 0.3 |
| Chills | 32/40 (80%) | 21/24 (87%) |
| Headache | 26/40 (65%) | 15/25 (60%) |
| Backache | 18/39 (46%) | 7/25 (28%) |
| Muscle ache | 19/39 (49%) | 10/25 (40%) |
| Fatigue | 34/40 (85%) | 23/25 (92%) |
| Joint pain | 13/39 (33%) | 7/25 (28%) |
| Antibiotic treatment | 39/48 (81%) | 22/30 (73%) |
| History of international travel | 13/46 (28%) | 10/30 (33%) |

Continuous data are presented as mean ± standard error of mean.
[a]Data missing on 5 PI-IBS patients.
[b]Data missing on 13 PI-IBS patients and 7 controls.
[c]Data missing on 24 PI-IBS patients and 13 controls.

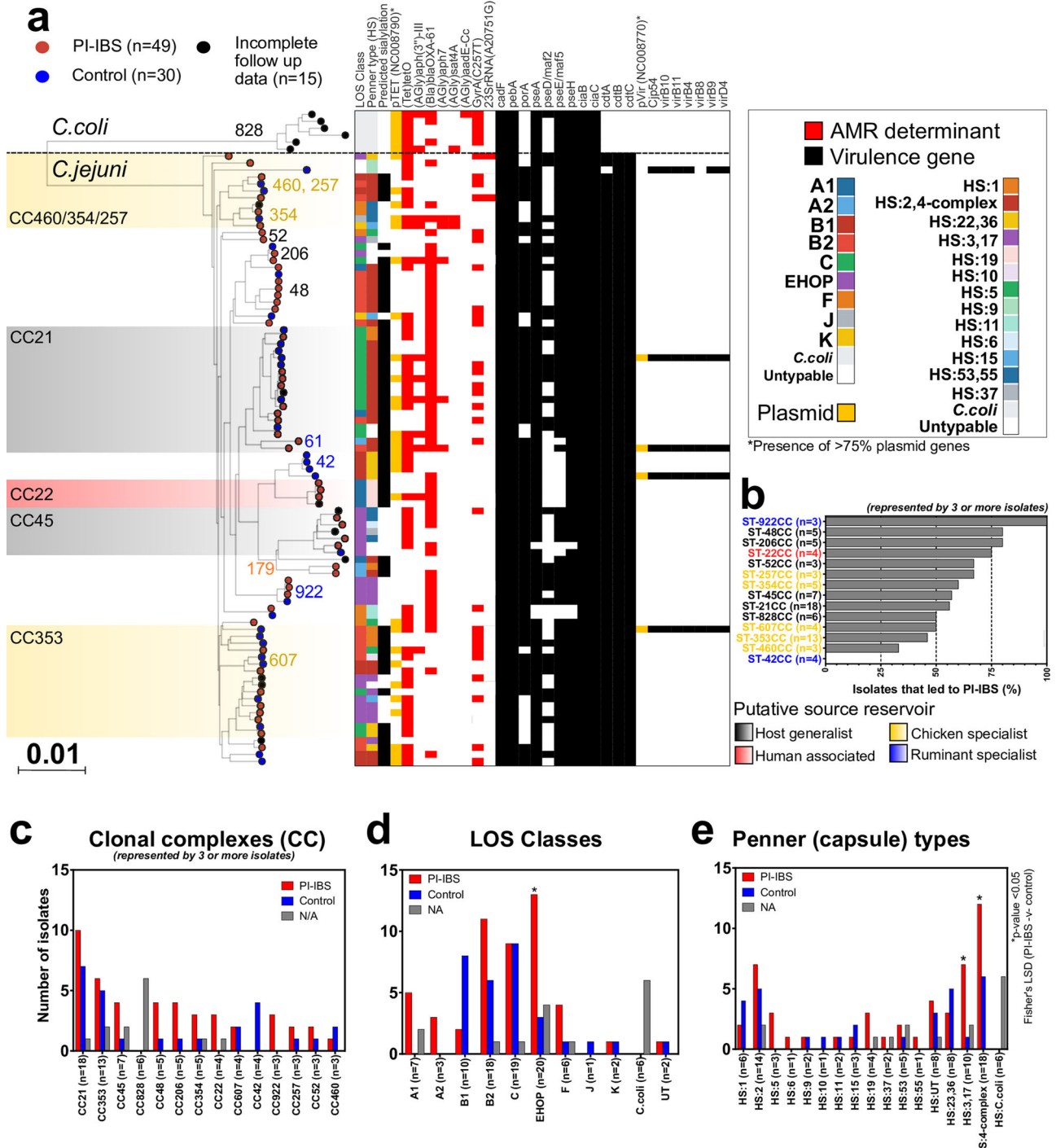

**Fig. 1 Population structure of *Campylobacter* isolates collected in this study. a** A maximum-likelihood phylogeny was constructed with IQ-TREE, using a GTR model and ultrafast bootstrapping (1000 bootstraps; version 1.6.8)[99,100] from an alignment of all isolates (*n* = 94). Scale bar represents a genetic distance of 0.01. Leaves from isolates that lead to an onset of PI-IBS are colored red and those that did not (control) are blue; isolates for which follow up date was incomplete are gray. The tree is annotated with MLST, LOS, and Penner (capsule) types, presence (colored block) and absence (no block) of common *Campylobacter* virulence factors (VFdB), antimicrobial resistance genes (ARG-ANNOT) and the presence of two previously characterized plasmids (pTet: NC008790 and pVir: NC008770; presence indicated if 75% of more plasmid genes present). Interactive visualization is available on Microreact[101]: https://microreact.org/project/CampyIBS. **b** Ratio of how many isolates from each MLST clonal complexes (CC) lead to the onset of PI-IBS. Only CCs represented by three or more isolates are shown. Frequency of **c** clonal complexes, **d** LOS classes and **e** Penner types identified. *p*-value < 0.05 indicated with (*); Fisher's LSD test. Raw data for (**b**–**e**) available in Supplementary Data 4.

cases; Fig. 1b, c), as were three quarters of the ST22 clonal complex (3 of 4) infection cases (Fig. 1b, c). The globally disseminated host generalist sequence types, ST48 and ST206 clonal complexes, also lead to the onset of PI-IBS in more than three

quarters of cases (4 of 5) (Fig. 1b, c). Conversely, neither acute infection with cattle associated ST42 clonal complex isolates (0 of 4), nor two thirds of cases involving the poultry-associated ST460 clonal complex, lead to the development of PI-IBS (1 of 3 led to

PI-IBS; Fig. 1b, c). Despite ST21 clonal complex being the most common ($n = 18$) lineage in acute infection, PI-IBS developed in just over half (55%) of cases. No significant difference in genome size was noted between *C. jejuni* isolates from PI-IBS cases and controls (Supplementary Fig. 1C). Despite differences between lineages, no source reservoir dominated; rather isolates from all ecological backgrounds were equally capable of leading to PI-IBS (Supplementary Fig. 1D).

**Association between lipooligosaccharide and capsule types and PI-IBS.** The lipooligosaccharide (LOS) and capsular polysaccharide (CPS) loci are highly variable in gene content[26–29] and this variability is reflected in the diversity of LOS and capsule types (Penner typing) for the PI-IBS and PI-no-IBS isolates ($n = 10$ LOS types; $n = 16$ capsule/Penner types; Fig. 1d, Supplementary Data 4). The most common LOS class locus was class B ($n = 10$ class B1; $n = 18$ class B2), which was evenly distributed between isolates that did and did not develop IBS. Most LOS class B isolates that developed IBS were of class B2 ($n = 11/18$). LOS class C was identified in 19 isolates, of which 9 (47%) developed PI-IBS. Three LOS classes showed some over-representation in PI-IBS isolates. LOS class EHOP was present in 20 isolates and 13/20 (65%) of these were from PI-IBS cases. LOS class F was present in six isolates and 4/6 (66%) of these were from PI-IBS cases. There were ten isolates with LOS class A and 80% developed PI-IBS symptoms (5 of 7 class A1; 3 of 3 class A2).

The most common CPS Penner type was HS:4-complex ($n = 18$) and exactly two thirds of these isolates were from PI-IBS cases ($n = 12/18$) and all were of LOS classes A and B (Fig. 1e, Supplementary Data 4). Additionally, Penner type HS:3,17 was present in ten isolates and 7/10 (70 %) of these were from PI-IBS cases (both Penner types statistically significantly overrepresented in PI-IBS). Penner HS:19 was present in four isolates, with three quarters from PI-IBS cases ($n = 3/4$). Some of the isolates that lacked complete follow up data (1 of 4 ST-22CC, 2 of 7 LOS class A1, and 1 of 4 Penner type HS:19) were also from these high risk lineages raising the question if they might be able to better predict PI-IBS development in larger datasets.

**Pathogenic *Campylobacter* have an open pangenome.** Construction of the pangenome using PIRATE on 88 *C. jejuni* and 6 *C. coli* isolates in our collection identified an open pangenome[30], meaning that the number of genes in the pangenome continues to increase with each additionally sequenced isolate (Supplementary Data 5). Accessory genes represented nearly three-quarters of the pangenome (3410 genes; 74% of pangenome; Supplementary Fig. 2) with a quarter of the genes identified (1225, 26%) considered core genes present in 95% or more of the isolates. A large proportion of the accessory genome, 2571/3,410 accessory genes (75.4%), was present in fewer than 15% of isolates. Differences in the distribution of pairwise genomic distances are indicative of multiple genetically distinct clusters that are diverging in both core sequences and accessory gene content. Visualization of this clustering revealed distinct *C. jejuni* and *C. coli* populations (Supplementary Fig. 1D). *C. jejuni* isolates clustered into three groups. One was composed primarily of isolates from poultry-associated clonal complexes (ST257, ST353, ST354, ST460, and ST607). Another was composed primarily of host generalist clonal complex isolates (ST21, ST52, and ST48). The third group consisted of isolates from the other large host generalist clonal complex ST21, the Guillain-Barre syndrome associated ST22 clonal complex[15], wild bird lineages (ST179 and ST692 clonal complexes) and ruminant-associated clonal complexes (ST61 and ST42; Supplementary Fig. 1D).

**Known virulence factors do not fully explain the onset of PI-IBS.** Virulence is often associated with variation in accessory genome content and antimicrobial resistance genes[7]. In our isolate collection, over half the isolates harbored resistance elements for tetracycline (63.8% *tetO*) and/or beta-lactamase (61.7% blaOXA-61); over a third (35.1%) harbored the fluoroquinolone resistance single nucleotide polymorphism (SNP), *gyrA* C257T; small numbers of isolates harbored aminoglycoside resistance elements [12.8% *aph(3')-III*; 5.3% *aph7*; 3.2% *sat4A*; 1.1% *aadE*]; and two isolates harbored a SNP in the 23S rRNA (A2075G) linked to erythromycin resistance (Fig. 1a; Supplementary Data 4). Tetracycline resistance in more than a third of the isolates (36.2%) was associated with a previously identified *C. jejuni* plasmid, pTet (accession: NC008790).

The presence of known virulence genes associated with *Campylobacter* adhesion, invasion, motility, toxin, and secretion system was also identified. Outer membrane proteins (*cadF*, *pebA*, *porA*), invasion proteins (*ciaBC*), glycosylation systems (N-linked, O-linked), motility (flagella genes), cytolethal distending toxin (*cdtABC*), lipooligosaccharide locus (LOS), and capsule production locus (CPS) were present in most isolates (74.5–100%), but only 4 isolates encoded a Type IV secretion system associated often found on the *C. jejuni* plasmid pVir (accession: NC008770). We did not observe significant differences in the frequency of virulence factors and antibiotic resistance genes between isolates from the PI-IBS cases and controls (Fig. 1a, Supplementary Data 4).

**Pangenome-wide association study of PI-IBS-associated genes.** A series of customized bacterial pGWAS were performed to investigate variation in the core and accessory genome of all *C. jejuni* isolates with PI-IBS follow-up data ($n = 79$). *C. coli* genomes ($n = 6$) were not included in the GWAS in order to minimize the confounding effects of bacterial lineages on identification of significant associations. Genomes from isolates that elicited PI-IBS ($n = 49$) were compared to control isolates ($n = 30$) where follow-up data indicated no disease sequelae. Input matrices of core SNPs, gene presence, allelic variation, and gene fusions and duplication were prepared following removal of low frequency alleles/genes (present in <0.05% of isolates) and SNPs (present in <0.1% of isolates).

Three correlation scores were employed (terminal, simultaneous, and subsequent) in order to differentiate between within- and between-lineage associations (Supplementary Data 6). In total, we identified variation in 24,528 genetic elements associated with PI-IBS (Table 2; Bonferroni corrected $p$-value $\leq 0.05$): 22,686 SNPs, 1392 alleles and 211 genes and 239 gene duplication, fissions or fusions. The subsequent score incorporates a clonal frame phylogeny that accounts for the impact of recombination (Fig. 2a; Supplementary Data 7), reducing the effect of population structure and maximizing the chance of identifying elements associated with the onset of PI-IBS. In total, 6311 subsequent associations were identified with a Bonferroni corrected $p$-value below 0.05: 5,896 SNPs; 335 alleles, 35 genes and 45 gene duplication, fissions or fusions, representing less than 4% of the total variants investigated (Fig. 2b). Associated variants were mapped back to the PIRATE pangenome file, which incorporated the NCTC 11168 reference genome to allow comparison with established gene nomenclature and help identify putative gene function. The variants that were most significantly associated with PI-IBS ($p$-value $<7.5 \times 10^{-5}$) were in genes linked with bacterial stress response (*Cj0145_phoX*) and several genes involved in core biosynthetic pathways (biotin: *Cj0308_bioD*; purine: *Cj0514_purQ* [x2]; and isoprenoid: *Cj0894c_ispH*; Fig. 2a).

**Table 2 Summary of all associated GWAS variants (p-value < 0.05), including terminal, simultaneous and subsequent scores from treeWAS[110].**

| Association Type | Total hits (p < 0.05) | SNP | Accessory gene presence | Accessory allele presence | Gene duplication | Gene fission/fusion | Core allele presence | Negatively associated | Positively associated |
|---|---|---|---|---|---|---|---|---|---|
| Terminal | 8090 | 7397 | 96 | 327 | 12 | 59 | 199 | 5537 | 2553 |
| Simultaneous | 10,127 | 9393 | 80 | 287 | 46 | 77 | 244 | 5309 | 4818 |
| Subsequent | 6311 | 5896 | 35 | 163 | 2 | 43 | 172 | 4748 | 1563 |
| Total | 24,528 | 22,686 | 211 | 777 | 60 | 179 | 615 | 15,594 | 8934 |

*Negatively associated GWAS hits (p value < 0.05)*

| Association Type | Total hits (p < 0.05) | SNP | Accessory gene presence | Accessory allele presence | Gene duplication | Gene fission/fusion | Core allele presence |
|---|---|---|---|---|---|---|---|
| Terminal | 5537 | 4904 | 93 | 322 | 12 | 59 | 147 |
| Simultaneous | 5309 | 4909 | 51 | 169 | 18 | 50 | 112 |
| Subsequent | 4748 | 4387 | 34 | 161 | 2 | 43 | 121 |

*Positively associated GWAS hits (p value < 0.05)*

| Association Type | Total hits (p < 0.05) | SNP | Accessory gene presence | Accessory allele presence | Gene duplication | Gene fission/fusion | Core allele presence |
|---|---|---|---|---|---|---|---|
| Terminal | 2553 | 2493 | 3 | 5 | 0 | 0 | 52 |
| Simultaneous | 4818 | 4484 | 29 | 118 | 28 | 27 | 132 |
| Subsequent | 1563 | 1509 | 1 | 2 | 0 | 0 | 51 |

In addition to identifying variants with the strongest association with PI-IBS, we identified hot spots of associated variants (Fig. 2c). Large numbers associated variants (multiplied terminal, simultaneous and subsequent p-value below 0.000125; Fig. 2c) were found in *Cj0208* (DNA modification methylase), *Cj0261c* (transducer-like, methyl-accepting chemotaxis proteins), *Cj0533* (succinyl-CoA ligase; biosynthesis of secondary metabolites), *Cj0720_flaC* (flagellar formation), the O-linked glycosylation locus (*Cj1293, Cj1300, Cj1334_maf3, Cj1351_pldA*; flagellar glycosylation), *Cj1414_kpsC* (capsule biosynthesis), *Cj1729c_flgE2* (flagellar secretion) and accessory genome gene cluster g00183 (hypothetical protein). Associated elements were predominantly core genome SNPs (Fig. 2d) with the most stringently associated elements found in multiple lineages (Fig. 2a).

**Risk-score provided varying predictive accuracy for different lineages.** The 1000 most associated GWAS elements (subsequent *p*-value below 0.0075) were tested for their ability to predict the onset of PI-IBS and their suitability as markers of infection risk. Twenty-two markers were able to accurately predict the onset of PI-IBS in more than 60% of cases (Fig. 3a). Four of the most accurate predictor variants were found in a single gene, *Cj1631c* a putative heme oxidase linked to virulence[31] (Table 3). Two predictors were identified in genes that contained many associated variants (flagellin C; *Cj0720c_flaC* and phospholipase A; *Cj1351_pldA*). By weighting each marker by its subsequent association score, which incorporates the direction of association, presence in multiple lineages, and strength of association, we used these 22 markers to calculate a cumulative score for each isolate and estimate the risk of developing PI-IBS (Fig. 3b).

Reflecting our empirical clinical evidence, we observed variation in risk between clonal complexes represented by 3 or more isolates (Fig. 1c, Supplementary Data 8). The highest risk was posed by isolates from ST48 ($n = 5$), ST45 ($n = 5$), ST206 ($n = 5$), ST22 ($n = 3$), and ST922 ($n = 3$) clonal complexes (mean risk score 1593). The ruminant associated ST42 clonal complex ($n = 4$) was predicted to be the least likely to lead to PI-IBS, with the lowest mean risk score ($-1344$). Risk prediction was also influenced by lineage, with accuracy varying from 67% in ST460 and ST257 clonal complexes to 100% accuracy in clonal complexes ST922, ST42, ST22, and ST52 (Fig. 3b, c). Repartition of the risk scores according to LOS class and Penner types also identified LOS classes A1 (1593) and EHOP (1584); and Penner types HS:19 (1593), HS:3,17 (1575) as high-risk lineages (Fig. 3d, e). Our clinical data suggested that Penner type HS:4-complex was also high-risk (Fig. 1e), but variability in isolate scores for this well-represented group ($n = 18$) lowered group's mean risk score and predictive accuracy (1415).

**PI-IBS isolates are more adhesive and invasive on colonocytes.** A random set of PI-IBS and control isolates ($n = 28$/group; each isolate tested in biological and technical replicates; indicated in Supplementary Data 9) was used for determining in vitro virulence. Adhesion was found to be significantly higher with PI-IBS ($2.19 \times 10^{-3} \pm 1.71 \times 10^{-3}$) compared to control strains ($1.75 \times 10^{-3} \pm 2.68 \times 10^{-3}$), Mann–Whitney, *p* value = 0.0025 (Fig. 4a). Invasion was also found to be significantly higher with PI-IBS strains ($5.65 \times 10^{-4} \pm 1.77 \times 10^{-3}$) compared to control strains ($1.55 \times 10^{-4} \pm 2.33 \times 10^{-4}$), *p* value = 0.005 (Fig. 4b). Results are expressed as (mean ± SD). Adhesion and invasion showed strong positive correlation (Spearman correlation, $r = 0.87$, $p < 0.0001$) (Fig. 4c). TER drop (% from baseline) was not significantly different between PI-IBS and control strains (PI-IBS: 90.07% ± 7.98 vs. control: 83.79% ± 16.05, $p = 0.058$) at 24 h post-infection (Fig. 4d).

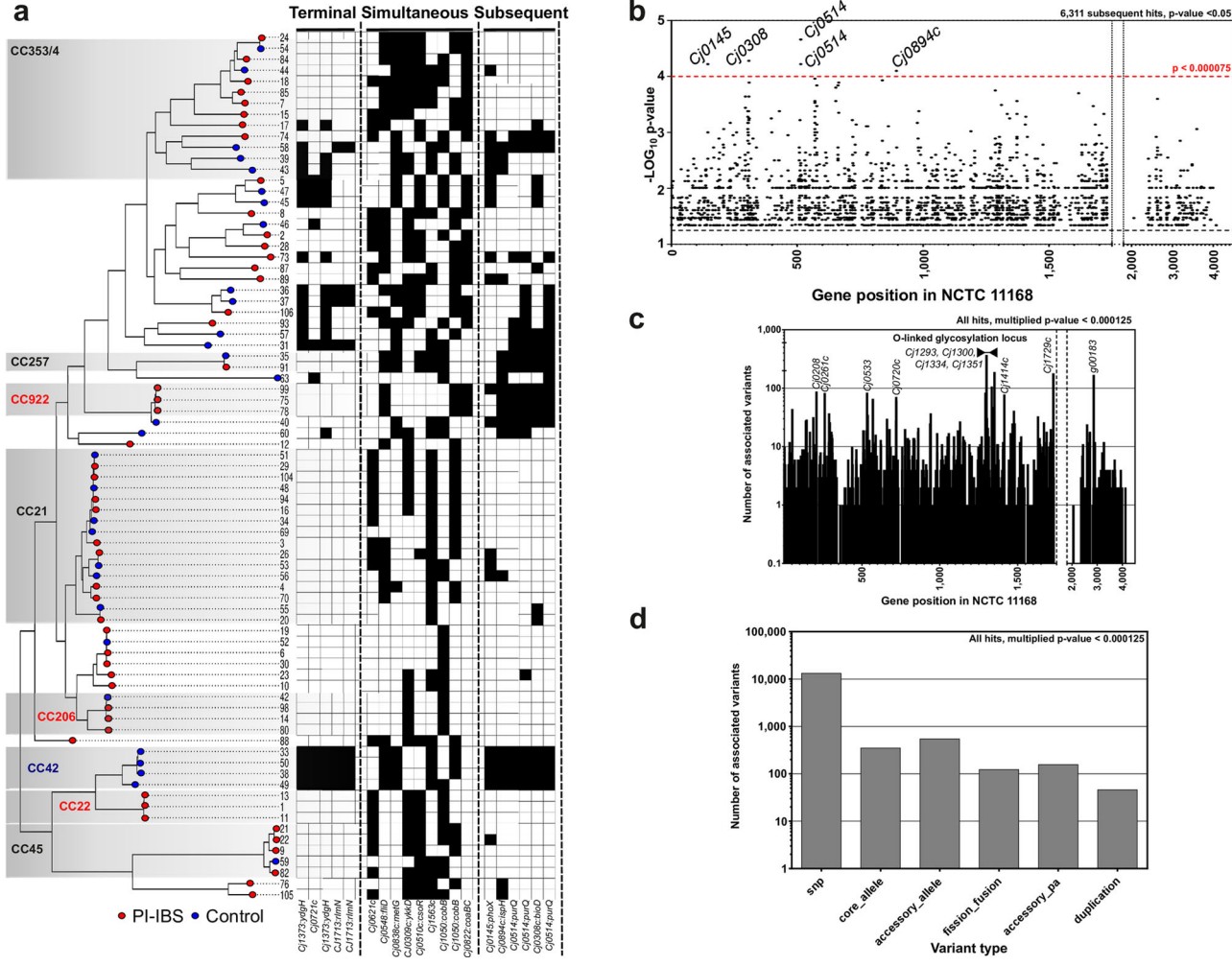

**Fig. 2 Genome-wide association of genetic variants with onset of PI-IBS. a** ClonalFrame-ML phylogeny of PI-IBS isolates, with recombination masked[114]. Red leaves are isolates from PI-IBS patients and leaves with blue circles are from control patients. Common disease-associated clonal complexes (CCs) are highlighted. Interactive visualization is available on Microreact[101]: https://microreact.org/project/CampyIBS-CF. Matrix shows presence of variants most associated with IBS, which are labeled below (terminal, $n = 5$; simultaneous, $n = 9$; subsequent, $n = 6$). **b** Pangenome position of associated GWAS results (subsequent $p$-value < 0.05; $n = 6,311$; Table S4). Each circle represents a variant mapped to a position in the *C. jejuni* reference strain NCTC11168[56] or inferred pangenome from this study (Supplementary Data 5). Variants with the strongest association ($p$-value < 0.000075; $n = 6$) from four genes with homologs in NCTC11168 are annotated (*Cj0145:phoX*, *Cj0308:bioD*, *Cj0514:purQ*, and *Cj0894c:ispH*). **c** Hot spots of associated variants. The number of associated variants per gene in NCTC11168 and the pangenome (any association $p$-value < 0.05). Genes with more than 100 associated variants are annotated, including four from the O-linked glycosylation locus. **d** Breakdown of associated variant types (core genome SNPs, core and accessory alleles, gene fission/fusions, accessory gene presence, and gene duplications; any association $p$-value < 0.05, raw data are available in Supplementary Data 6).

**Variation in accessory genes is associated with in vitro virulence.** Endonuclease (*Cj0139*), *Campylobacter* adhesion protein *capA* (*Cj0628*), O-linked glycosylation (*Cj1293-Cj1342*) and glycan transferase (*Cj1438*) represented the greatest number of unique alleles per isolate (Fig. 5a). Prevalence of core gene variants acetyltransferase (*Cj0295*), and the genes with uncharacterized/unknown function (*Cj0567*, *Cj0569*, *Cj0570*) were higher in PI-IBS while *Cj1392* was higher in control isolates (Fig. 5b). Adhesion was associated with *capA* (*Cj0628*), periplasmic protein (*Cj0967*), β-1,3 galactosyltransferase (*Cj1139*) and O-linked glycosylation (*Cj1293-Cj1342*) (Fig. 5c). Invasion was also associated with *capA* (*Cj0628*) as well as lipo-oligosaccharide locus (*Cj1145*) genes (Fig. 5d). One variant (*Cj0139*) was associated with TER drop (Fig. 5e). Increased variation in accessory genes was found to be significantly associated with adhesion (Fig. 5f, Supplementary Data 10). However, there were no overall differences in core, accessory, GWAS hit and O-linked glycosylation genes for invasion and barrier disruption (Fig. 5f).

**Chemokine and cytokine induction by *Campylobacter* isolates.** Apical media fractions from T84 cells exposed to PI-IBS associated *C. jejuni* strains had significantly higher IL-8 as compared to media fractions from control strains (PI-IBS: 117.7 (27.38) pg/mL vs. control: 97.8 (27.85) pg/mL, $p = 0.01$) (Fig. 6a). Basolateral concentrations of IL-8 were not different between the strains (Supplementary Fig. 3A). Apical media fractions from T84 cells exposed to PI-IBS associated *C. jejuni* strains showed lower chemokine (C–C motif) ligand 2 (CCL2) or monocyte chemoattractant protein-1 concentrations compared to the media fractions from cells exposed to control strains (PI-IBS: 1.81 (0.96) pg/mL vs. control: 2.84 (1.49), $p = 0.004$) (Fig. 6b), while basolateral concentrations were not different (Supplementary Fig. 3B). Basolateral TNFα concentrations were higher after exposure to PI-IBS strains compared to control strains (PI-IBS: 0.30 (0.37) pg/mL vs. control: 0 (0) pg/mL, $p = 0.0001$) (Fig. 6c), as well as C-X-C motif chemokine ligand 10 (CXCL10) or IP-10 (PI-IBS: 6.81 (7.01) pg/mL, control: 2.72 (4.64) pg/mL, $p = 0.01$)

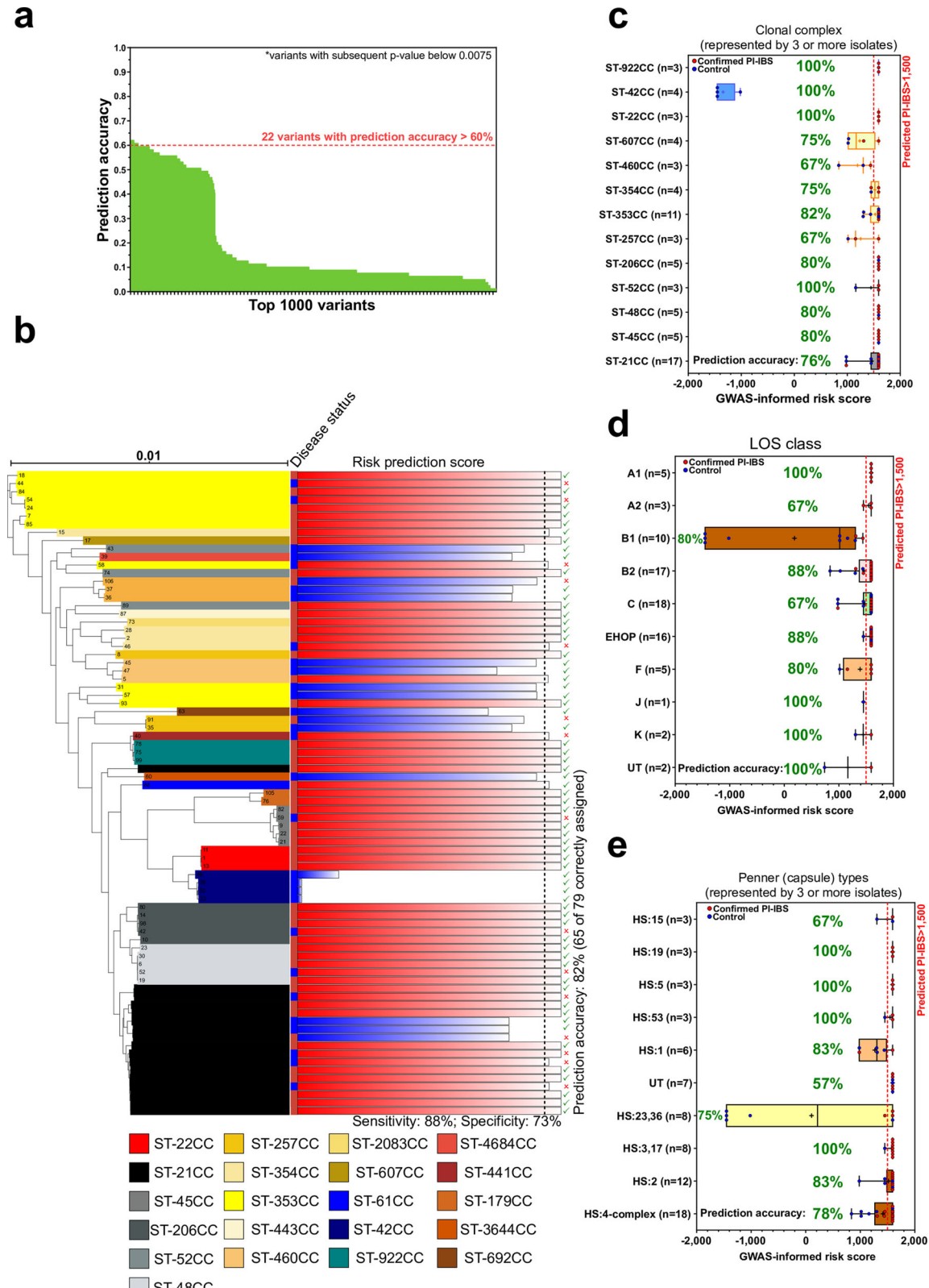

(Fig. 6d), while apical concentrations of either were not different (Supplementary Fig. 3C, D). The IL-1α (apical), IL-6 (apical), MIG (apical or basolateral), and I-TAC (apical or basolateral), levels were unchanged in response to the two bacterial groups ($p > 0.05$) (Supplementary Fig. 3E–J). IFNγ, IL-10, IL-1β, and IL-4 levels (apical or basolateral) and IL-1α, IL-6 (basolateral) were not within the readable range upon exposure to any *C. jejuni* strain.

**High risk lineages elicit greater apical IL-8 and basolateral TNFa induction**. Comparison of cytokine responses within and between lineages highlighted that the response of each individual isolate was not similar across all cytokines (Fig. 6e). Variation was greatest in response to apical IL-8 and basolateral TNFα and significant differences (ANOVAs with Tukey correction for multiple comparisons) were observed in response to several of

**Fig. 3 Predicting risk of PI-IBS using risk markers derived from the primary GWAS. a** Ability of top 1,000 (of 6,311) variants from the GWAS to predict the onset of PI-IBS (variants with subsequent *p*-value below 0.0075). **b** ClonalFrame-ML phylogeny of IBS isolates, with recombination masked. IBS onset is denoted by red (PI-IBS) and blue (control) squares. Common disease-associated clonal complexes (CCs) are highlighted. Predicted risk score indicated by the red or blue bars (color denoting predicted outcome). High risk isolates scored above 1500 and small tick (✔) or cross (✗) indicated whether the prediction observed disease status. **c** Repartition of risk scores on 83 isolates used in the GWAS from acute *C. jejuni* infection, according to the onset of PI-IBS within 9 months. Each point corresponds to the risk score associated with a single isolate. Risk score was calculated by weighting the association score of the top 22 PI-IBS predictive variants (Table 2). PI-IBS was predicted with a score above 1500, which showed variation between lineages (disease outcome was correctly predicted in 65 of 83 isolates). **d** Repartition of risk-score prediction by LOS class and **e** Penner types. *All box plots show the 25th and 75th percentile, whiskers extending to the min and max values with all data points shown. Raw data for (**b**–**e**) are available in Supplementary Data 9.

**Table 3 Details of GWAS-informed markers (n = 22) used to predict risk of subsequent PI-IBS.**

|   | Variant | gene.family (PIRATE) | Locus ID (NCTC11168) | gene.name | Prediction accuracy |
|---|---------|----------------------|----------------------|-----------|---------------------|
|   | 203498_C_C-core | g00050 | Cj0112 | tolB | 60.8% |
|   | 1123058_C_C-core | g01209 | Cj0118 | ttcA | 60.8% |
|   | 816843_T_T-core | g02780 | Cj0231c | nrdB | 60.8% |
|   | 690709_G_G-core | g01096 | Cj0289c | NA | 60.8% |
| # | 622830_C_C-core | g00792 | Cj0513 | purS | 60.8% |
| * | 859191_C_C-core | g00505 | Cj0720c | flaC | 60.8% |
|   | 1267111_G_G-core | g00879 | Cj0862c | pabB | 60.8% |
|   | 398511_A_A-core | g00018 | Cj1027c | gyrA | 60.8% |
|   | 1326123_T_T-core | g00773 | Cj1042c | exsA | 60.8% |
|   | 222728_C_C-core | g00083 | Cj1120c | pglF | 60.8% |
|   | 940026_T_T-core | g01168 | Cj1126c | pglB | 60.8% |
|   | 1159550_C_C-core | g00762 | Cj1199 | efe | 60.8% |
|   | 883365_A_A-core | g01037 | Cj1264c | hybD | 62.0% |
| * | 1265794_T_T-core | g00380 | Cj1351 | pldA | 62.0% |
|   | 1074024_G_G-core | g03292 | Cj1411c | ptlI | 62.0% |
|   | 431464_G_G-core | g00550 | Cj1412c | NA | 62.0% |
|   | 469831_G_G-core | g00386 | Cj1631c | NA | 62.0% |
|   | 469836_A_A-core | g00386 | Cj1631c | NA | 62.0% |
|   | 470282_A_A-core | g00386 | Cj1631c | NA | 62.0% |
|   | 469774_G_G-core | g00386 | Cj1631c | NA | 62.0% |
|   | 597240_G_G-core | g01291 | Cj1641 | murE | 62.0% |
|   | g01925_00012-c | g01925 | Cj1711c | rsmA | 62.0% |

#Among most associated GWAS hits.
*Gene with high number of GWAS hits.

our clinically observed/GWAS predicted high risk lineages. Apical IL-8 responses for isolates from ST-206 and ST-22 clonal complexes, LOS class A1 and Penner type HS:19 were significantly higher than other tested lineages. Similarly, the same lineages (apart from ST-206 CC) elicited increased basolateral TNFα production (Fig. 6e). However for most cytokines, PI-IBS status was the more significant variable in the analysis despite variation in cytokine induction between lineages (Supplementary Fig. S3).

## Discussion

The epidemiology of PI-IBS is well established but understanding the pathophysiology remains an obstacle to effective interventions[6]. Intestinal barrier dysfunction[32–36], microbial dysbiosis[33,37–39], and immune dysregulation[32,40] have all been described in PI-IBS patients, and the intensity of acute illness is linked to the development of PI-IBS[6]. Host factors are undoubtedly important as evidenced by increased risk observed in younger people and women after enteric infection. However, the possibility that some strains are more virulent, and, therefore, more likely to cause the onset of PI-IBS has remained unexplored.

Using comparative genomics techniques and in vitro studies, we identified pathogen factors associated with the onset of PI-IBS at multiple levels. First, we examined variation in the strains associated with the development of PI-IBS to identify high and

low risk lineages. Second, we recorded the difference in presence/absence patterns of known *Campylobacter* virulence and pathogenicity genes among patients who develop PI-IBS and those who do not. Third, we conducted a GWAS approach that was agnostic to existing assumptions about gene function and virulence to identify genetic variation in genes that was associated with the development of PI-IBS. Finally, we investigated clinically relevant *C. jejuni* phenotypes in vitro to characterize variation in adhesion, invasion, barrier function disruption, and induction of pro-inflammatory cytokine production among PI-IBS and control *C. jejuni* isolates.

The most common isolates obtained from patients with PI-IBS belonged to the ST21 clonal complex. This lineage is among the most common causes of acute human infection[41,42] and has been associated with extra-intestinal spread and liver infection[43]. However, the prevalence of ST21 clonal complex isolates was approximately equal among PI-IBS cases and controls. A more striking finding was the differential association observed in other strains. Specifically, isolates belonging to the ST22 clonal complex were over-represented in isolates that lead to the development of PI-IBS, while ST42 clonal complex were very low risk. Interestingly, while ST22 complex isolates are a relatively infrequent cause of gastroenteritis, they account for up to a third of infections among patients who developed Guillain-Barre syndrome following campylobacteriosis[15,44]. It is known that the structure

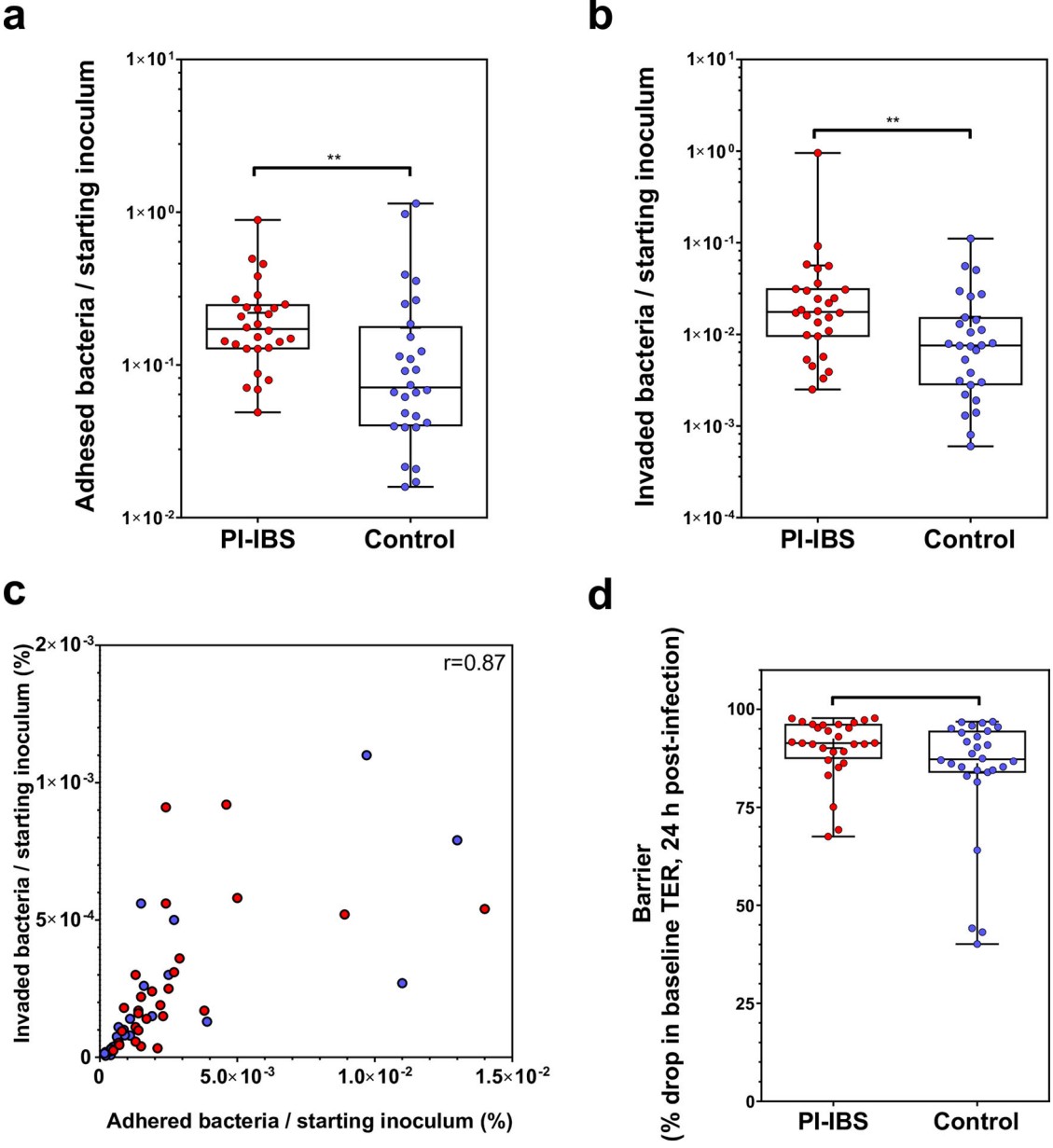

**Fig. 4 In vitro pathogenicity phenotypes of PI-IBS and control *C. jejuni* isolates. a** PI-IBS causing *C. jejuni* isolates demonstrate greater adhesion to T84 epithelial cells. **b** Using gentamycin protection assay, PI-IBS isolates demonstrate greater intracellular invasion compared to the control isolates. **c** Positive correlation between adhered and invaded bacteria on T84 colonocytes (non-parametric Spearman correlation, $r = 0.87$; *p*-value < 0.0001). **d** T84 colonocytes were grown as epithelial monolayers with transepithelial resistance (TER) > 750 Ω cm². No differences in % drop in TER in monolayers exposed to PI-IBS causing *C. jejuni* isolates compared to control isolates. ($n = 28$/group, Mann–Whitney *U* test). *All box plots show the 25th and 75th percentile, whiskers extending to the min and max values with all data points shown. Raw data are available in Supplementary Data 9.

of *C. jejuni* lipooligosaccharide (LOS) has a role in the outcome of infection. Specifically, isolates that can produce sialylated ganglioside mimics[45], LOS (class A, B, or C), have greater potential for invasion and translocation across epithelial cell monolayers[46–48] and are more frequently associated with bloody diarrhea and a longer duration of symptoms[49,50]. All class A isolates in our study lead to development of PI-IBS, although class B and C isolates were found among control and PI-IBS isolates. While this provides a possible explanation for the PI-IBS association of ST22 clonal complex isolates[51]; there was no clear association of sialylated LOS classes with PI-IBS, as exhibited in Guillain-Barre syndrome and Miller Fisher Syndrome. Lastly, Penner (capsule)

HS:3,17 and HS:4 complex were significantly more common in PI-IBS isolates.

The most obvious place to start is to consider differential prevalence of putative virulence genes in *Campylobacter*. However, while many have been described[46–49], there is limited data on the contribution of these virulence genes to variability in acute disease severity or the risk of post-infection sequelae. Furthermore, as virulence genes are typically those that are common in isolates from infected patients, it is no surprise that the majority of *Campylobacter* from human infection studies possess these common genes[52]. Consistent with this, we found no significant difference among PI-IBS cases and controls in the prevalence of

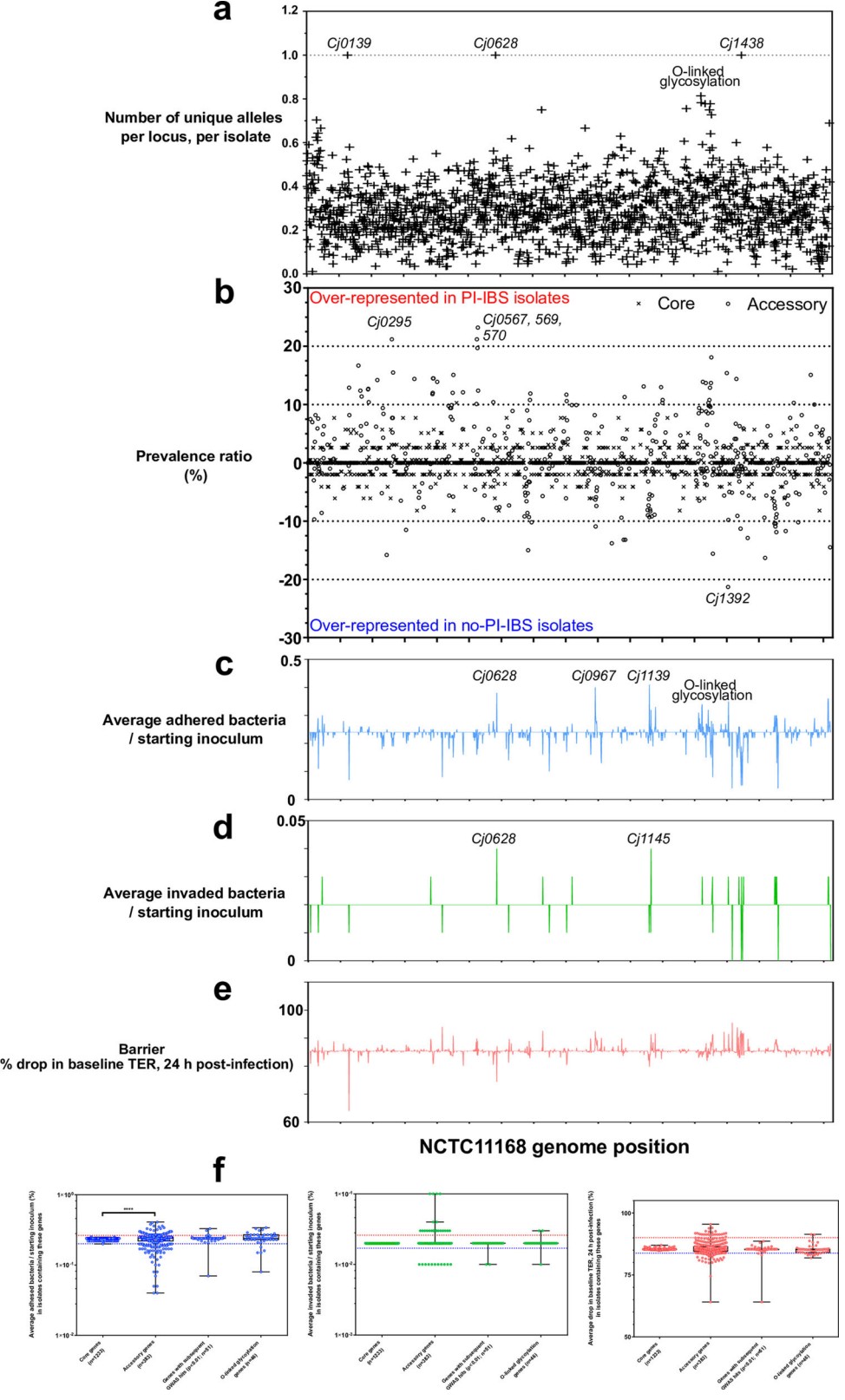

common virulence genes including outer membrane proteins (*cadF*, *peb1*, *porA*); invasion proteins (*ciaB*); motility (flagella genes); cytolethal distending toxin (*cdtA-C*. It is, therefore, necessary to take other approaches to investigate genomic differences between the strains.

GWAS analysis revealed variation in gene presence, core and accessory gene alleles, core SNPs, gene fission/fusions, and gene duplications. Five variants were most significantly associated with PI-IBS (Bonferroni corrected $p$-value $< 7.5 \times 10^{-5}$). These included genes associated with bacterial stress response (*Cj0145_phoA*), core biosynthetic pathways (biotin: *Cj0308_bioD*; two variants in purine: *Cj0514_purQ*; and isoprenoid: *Cj0894c_ispH*). Iron, though important for several physiological processes can be toxic. Thus, several Gram-negative bacteria control cytoplasmic iron

**Fig. 5 Effect of genome-wide variation on in vitro pathogenicity-related phenotypes. a** Allelic diversity of NCTC 11168 homologs in 83 isolates used in the GWAS. **b** Gene prevalence of all NCTC 11168 genes. Genes over-represented in PI-IBS or control isolates appear further away from the mid-point. **c** Correlation of gene presence with pathogenicity-related phenotypes. Average number of adhered bacteria as a ratio to starting inoculum when NCTC 11168 genes are present in an isolate. **d** Average number of invaded bacteria as a ratio to starting inoculum when NCTC 11168 genes are present in an isolate. **e** Average drop in baseline TER when NCTC 11168 genes are present in an isolate. **f** Summary of variation in in vitro pathogenicity phenotypes grouped by core (present in ≥95% of isolates) and accessory genes. Significantly more variation (t-test, Mann–Whitley, p-value = 0.0012) was attributed to the accessory genome in adhesion assays (blue), than invasion assays (green) or barrier integrity assays (red). Raw data are available in Supplementary Data 10. *All box plots show the 25th and 75th percentile, whiskers extending to the min and max values with all data points shown.

levels through the transcriptional regulator Fur[53]. Iron and Fur genes alter production of transducer-like proteins (tlp) that allow *C. jejuni* to sense environmental signals and aid bacterial invasion[54]. The *Cj0145* (*phoA*) gene is located immediately downstream of *tlp2* and induced in the presence of iron, likely regulating cytoplasmic iron levels[53]. It also shows homology with *Vibrio cholerae* alkaline phosphatase and plays a role in phosphate regulated two-component system. Inactivation of *Cj0145* resulted in reduced alkaline phosphatase activity, which regulates PhosS/PhosR mediated regulation of intracellular phosphate and survival in less favorable habitats[55]. ATP-dependent dethiobiotin synthetase *Cj0308* (*bioD*) catalyzes the first step in biotin biosynthesis from 7,8-diaminononanoate[56]. However, not much is known about the biological importance of biotin synthesis in *Campylobacter* pathogenesis. *Cj0514* (*purQ*) is part of the *pur* operon required for purine metabolism. The product (phosphoribosylformylglycinamidine synthase subunit I) catalysis ATP and glutamine dependent formation of formyl glycinamidine ribonucleotide, ADP, P (i), and glutamate in the de novo process of purine biosynthesis[57]. *Cj0894* (*ispH*) is involved in step 6 of the sub pathway that synthesizes isopentenyl diphosphate from 1-deoxy-D-xylulose 5-phosphate[56]. It is plausible that survival in a stressed environment provides the isolates greater opportunity for mediating epithelial injury and immune responses. However, the mechanistic interaction between these core biosynthetic genes and PI-IBS development would need further studies.

Eleven genes (*Cj0208_fokIM*, *Cj0261c_ubiG*, *Cj0533_sucC*, *Cj0720c_flaC*, *Cj1293_pseB*, *Cj1300_ubiG*, *Cj1334*, *Cj1351_pldA*, *Cj1414c_kpsC*, *Cj1729c_flgE*, *g00183*) had the greatest number of variants associated with PI-IBS. Three of these are involved in energy metabolism; *Cj0208*, a DNA modification methylase; *Cj0261c*, a ubiquinone biosynthesis O-methyltransferase; and *Cj0533*, an ADP-forming Succinate–CoA ligase subunit beta. *Cj0720* (*flaC*) encodes for non-structural flagellin which is secreted through the flagella and has been implicated in bacterial invasion[58]. Interestingly, the *flaC* showed significant sequence identity with TLR5-activating flagellins of other bacteria, like *Salmonella*, but not with other *Campylobacter* flagellins. In vitro studies showed direct interaction of *flaC* with TLR5 and p38 activation[59]. The immunomodulatory property of this protein may play a role in pathogenesis of PI-IBS. UDP-N-acetylglucosamine 5-inverting 4,6-dehydratase (*Cj1293_PseB*) catalyzes the first step in the biosynthesis of pseudaminic acid, which mediates flagellin glycosylation[60]. This has been shown to facilitate bacterial motility[61]. *Cj1334* is an uncharacterized gene but shares 90% homology with bacterial motility associated factor glycosyltransferase family protein. *Cj1351* (*pldA*) cleaves phospholipids resulting in lysophospholipids that contain only acyl chain instead of two[62]. These lysophospholipids accumulate in conditions of environmental stress and mediates survival in stressed conditions[63]. KpsC (*Cj1414c*), KpsM, and KpsS mutants lacked the ability to produce O antigen or the capsular polysaccharide[64]. kpsM mutant of strain 81–176 showed a 10-fold reduction in intestinal epithelial cells invasion in vitro[65]. *flgE* encodes for flagellar hook, which is a joint between the motor and

flagellar filament, and is found across a number of motile bacterial species[66,67]. Thus, in addition to stress response genes, variants in flagellar- and motility-associated genes are significantly associated with PI-IBS.

We found that both adhesion and invasion associated with CapA, a surface-exposed autotransporter protein that mediates invasion into the intestinal epithelial cells[68]. It was found to be the only conserved putative adhesins in a study of 97 *C. jejuni* isolates using dot blot hybridization and in vitro adhesion assays[69]. Adhesion was also associated with periplasmic protein *Cj0967*, which been shown to play a role in polysaccharide production, and biofilm formation, which promotes virulence and stress survival[70]. In NCTC 11168, WlaN or CgtB (*Cj1139*) demonstrated β-1,3 galactosyltransferase activity which mediates biosynthesis of the LOS core oligosaccharide[71,72]. We found *Cj1139* to be associated with adhesion. Another closely related LOS locus and homopolymeric G/C tract containing gene (*Cj1145*) is associated with invasion in our study which has been reported in the past[72]. *Cj0139* is a putative endonuclease whose role was investigated in heat shock mediated conjugation efficiency, an important process to horizontal gene transfer[73]. However, its inactivation did not result in conjugation efficiency in heat shocked or untreated cells. In our study, its variants were associated with disruption of epithelial barrier in vitro. Overall, we found that PI-IBS causing isolates displayed elevated in vitro virulence. Adhesion to epithelial cells, intracellular invasion and survival, elevated pro-inflammatory cytokine response and disruption to barrier function are all known to be associated with virulence in humans[74]. Invasion as well as CDT production plays a role in IL-8 secretion[75,76]. In addition to providing a route for its own translocation into the sub-epithelial space, invasion is known to facilitate translocation of commensal *E. coli*[77,78]. The PI-IBS causing isolates in our study were twice as invasive as the control isolates. In a recent study based on transcriptomics, genes associated with oxidative stress, N-, and O- glycosylation systems, translocases and the membrane-integrated component of the flagellar apparatus associated with bacterial internalization[79]. Bacterial motility induced by the flagella is critical for *Campylobacter* adhesion and invasion[80], and flagellar export apparatus also mediate intracellular secretion of other *Campylobacter* invasive antigen (Cia) proteins[19,81], which have been associated with facilitating intracellular survival[82,83]. *C. concisus* strains from patients with chronic inflammatory bowel disease had greater invasion potential and caused induction of IFN-γ[84]. However, PI-IBS differs in pathophysiology and patients do not demonstrate chronic colonization of *Campylobacter* species in the gastrointestinal tract.

Predicting the development of disease sequel not only increases understanding of pathogenesis but also has considerable potential to inform treatment and preventative interventions. Similar to other studies that have aimed to predict disease severity following infection by different bacterial genotypes, we used a GWAS-informed approach to select genetic markers for the onset of PI-IBS[85–89]. By weighting the presence of 22 PI-IBS associated markers according to their disease-association score, we were able

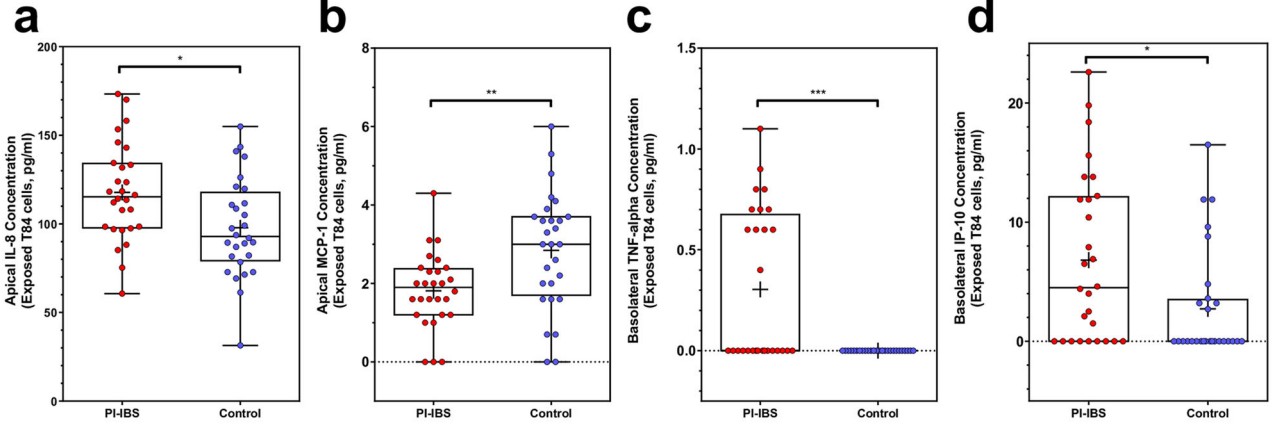

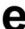

to correctly predict the long-term disease outcome in up to 78% of cases. Particularly, ST22 and 922 were universally associated with PI-IBS and ST42 with the control state. Our results show promise that an artificially enhanced disease outcome prediction model could be applied with large reference datasets, incorporating the breadth of bacterial species (and host) population structure. Challenges for risk prediction based upon infection associated markers include accurate diagnosis of disease sequelae and confounding variability of the host immune system. A more sophisticated approach may balance the association of disease markers identified through bacterial GWAS correlated with differences in the host genome[90,91]. Our prior study showed that longer time before starting antibiotic therapy may increase the risk of PI-IBS[3]. Additionally, PI-IBS development is associated

**Fig. 6 Cytokine responses in colonocytes following exposure to PI-IBS and control *C. jejuni* isolates.** T84 epithelial monolayers were exposed to *C. jejuni* isolates for 24 h followed by collection of media from apical and basolateral compartments. **a** Apical IL-8 concentration higher in PI-IBS *C. jejuni* exposed colonocytes than exposed to control isolates. **b** Apical hemokine (C–C motif) ligand 2 (CCL2) or monocyte chemoattractant protein 1 (MCP-1) concentration lower in PI-IBS *C. jejuni* than control isolates. **c** Basolateral TNF-alpha concentration higher in PI-IBS *C. jejuni* than control isolates. **d** Basolateral C-X-C motif chemokine ligand 10 (CXCL10) or interferon gamma-induced protein 10 (IP-10) concentration higher in PI-IBS *C. jejuni* than control isolates. **e** Differences in apical IL-8 concentration, apical MCP-1 concentration, basolateral TNF-alpha concentration and basolateral IP-10 concentration in response to *C. jejuni* infection grouped by clonal complexes, LOS class and Penner type. Red and blue bars highlight high and low risk lineages, respectively. ($n = 28$/group, Mann–Whitney $U$ test). *All box plots show the 25th and 75th percentile, whiskers extending to the min and max values with all data points shown. Raw data are available in Supplementary Data 9.

with reduced diversity of commensal microbiota which can be enhanced through a targeted set of commensals or fecal microbiota transplantation[33]. In future, a validated risk-score may allow targeted interventions to prevent establishment of chronic phenotype of PI-IBS.

In conclusion, acute epithelial injury to cellular and paracellular compartments can signal downstream changes in immune and neuromuscular apparatus that may lead to changes in motility and sensation, which may contribute to the development of PI-IBS. Additionally, a subset of patients may be at risk for long-term changes in luminal (microbiota, bile acids, proteases, etc.) and epithelial compartment (epigenetic changes in secretory, barrier and other signaling mechanisms), which may make them vulnerable for PI-IBS development. Different strains of *Campylobacter* have also been shown to have unique toxigenic effects in vitro. This study provides the systematic evaluation of bacterial genomics and in vitro virulence using isolates acquired from a prospectively identified cohort of patients developing *Campylobacter* PI-IBS. Identification of key genes associated with bacterial metabolism, stress response, motility and ganglioside mimics, in conjunction with greater adhesion and invasion, demonstrate a clear contribution of bacterial factors in the long-term sequel of *Campylobacter* enterocolitis. In combination with host epidemiological factors, these findings pave the way for identification of high-risk infections that can be amenable for preventative strategies. Furthermore, experimental studies testing these genetic elements in animal models using sensory-motor assessments will advance our understanding of PI-IBS pathophysiology.

## Methods

**Study participants and bacterial strains.** Patients with stool culture positive *Campylobacter* enteritis were surveyed 6–9 months post infection to determine the presence of PI-IBS using Rome III criteria[22]. Details on survey strategy and clinical metadata collected are published[3]. Demographic and clinical symptom data during acute campylobacteriosis was collected for each patient. The study was approved by the institutional review boards at the Minnesota Department of Health and Mayo Clinic. All participants provided informed consent for participation in the survey and for sharing of biospecimens.

*C. jejuni* isolates were cultured on *Campylobacter* sheep blood agar plates (Hardy Diagnostics, #A40) along with reference isolates NCTC 11168[56] and 81–176[92] that were chosen as control strains. *C. jejuni* were routinely grown in Müeller-Hinton (MH) broth (BD BBL™) or on agar at 37 °C in a tri-gas incubator (Thermo Scientific, Heracell™ 150i) under microaerobic conditions (85% N$_2$, 10% CO$_2$ and 5% O$_2$).

**DNA extraction and sequencing.** Genomic DNA of all isolates was extracted using Wizard genomic DNA purification kits (Promega), following the manufacturer's instructions. DNA quality was measured using a Nanodrop (Thermo Scientific, USA) and Qubit fluorometer (dsDNA HS) kit (Thermo Scientific, USA). Indexed Illumina sequencing libraries were prepared using Nextera XT DNA Library Preparation Kits (Illumina Inc., San Diego, CA) according to the manufacturer's protocols. Briefly, 1 ng genomic DNA of each sample was tagmented by Nextera transposome, and then libraries were amplified by barcoded primers. Library cleanup and size selection were performed to remove amplicons <300-bp using AMPure XP beads (NEB, USA). Libraries were normalized, pooled, and sequenced on Illumina HiSeq 3000 platform with 150-bp paired-end reads for all the isolates. Raw sequencing reads were quality trimmed and filtered using Trimmomatic (version 0.33)[93] and assembled using the de novo assembly

algorithm, Velvet (version 1.2.08). The VelvetOptimiser script (version 2.2.4) was run for all odd k-mer values from 21 to 99[94,95]. Read data were deposited in the Sequence Read Archive under BioProject ID PRJNA675124.

**Core and accessory genome characterization.** All unique genes present in at least one isolate (the pangenome) were identified by automated annotation using PROKKA (version 1.13) followed by PIRATE, a pangenomics tool that allows for orthologue gene clustering in bacteria. We defined genes in PIRATE using a wide range of amino acid percentage sequence identity thresholds for Markov Cluster algorithm (MCL) clustering (45, 50, 60, 70, 80, 90, 95, 98). Genes in the pangenome were ordered initially using the NCTC 11168 reference followed by the order defined in PIRATE based on gene synteny and frequency. Core and accessory genome variation analysis was performed as previously described[14,96]. Briefly, a matrix was produced summarizing the presence/absence and allelic diversity of every gene in the pangenome list. Core genes were defined as present in 95% of the genomes and accessory genes as present in at least one isolate (Supplementary Data 1). The pangenome was visualized using phandango, as a matrix of gene presence alongside a core genome phylogeny[97].

An alignment of all 94 *Campylobacter* isolates was constructed from concatenated gene sequences of all core genes (found in ≥95% isolates) using MAFFT (version 7)[98] on a gene-by-gene basis (size: 429,515 bp; Supplementary Data 2). Maximum-likelihood phylogenies were constructed for *C. jejuni* and *C. coli* using a GTR + I + G substitution model and ultra-fast bootstrapping (1000 bootstraps)[99] implemented in IQ-TREE (version 1.6.8)[100] and visualized on Microreact (https://microreact.org/project/CampyIBS)[101]. Pairwise core and accessory genome distances were compared using PopPunk (version 1.1.4), which uses pairwise nucleotide k-mer comparisons to distinguish shared sequence and gene content to identify divergence of the accessory genome in relation to the core genome. A two-component Gaussian mixture model was used to construct a network to define clusters (Components: 43; Density: 0.1059; Transitivity: 0.8716; Score: 0.7793)[102].

**Molecular typing and identification of virulence and antibiotic resistance genes.** MLST analysis and examinations of known virulence associated genes and antibiotic resistance genes were carried out using the SRST2 pipeline, which uses raw Illumina reads as input[103]. Capsule polysaccharide (CPS) and lipooligosaccharide (LOS) locus types of each *C. jejuni* isolate were also characterized from their raw sequence data: short read sequences were mapped to known capsule and LOS locus types using BLAST as previously described[27,29,104,105]. Antimicrobial resistance genes and putative virulence genes were detected through comparison with reference nucleotide sequences in the VFDB[106] and ARG-ANNOT[107] databases, respectively. Point mutations related to antibiotic resistance genes were identified by PointFinder using the pointfinder_db database[108].

**Pangenome wide association studies (pGWAS).** GWAS analyses were used to compare PI-IBS cases confirmed by follow-up ($n = 49$) with isolates that did not elicit IBS symptoms 6–9 months following initial infection ($n = 30$). Variation in core and accessory genome elements were identified using PIRATE[109] and incorporated into treeWAS[110] to investigate associations in [i] gene presence, [ii] core and [iii] accessory allelic variants, [iv] core SNPs, [v] gene fission/fusions and [vi] gene duplications. We removed any alleles/genes with a frequency of <0.05% and any SNPs with a frequency <0.1%. Three scores were calculated in order to differentiate between within- and between-lineage associations: (1) the terminal score is phylogenetically naïve and measures sample-wide association across the leaves of the tree; (2) the simultaneous score identifies within-lineage association by measuring the parallel change in genotype and phenotype across the tree; and (3) the subsequent score incorporates the phylogeny to measure the proportion of the tree in which genotype and phenotype co-exist. The subsequent score is more sensitive and should be able to detect subtle patterns of association, while pooling the scores provides a comprehensive description of association in the dataset.

A *C. jejuni* specific phylogeny was constructed of the 79 *C. jejuni* isolates with complete IBS follow-up data by mapping assemblies against the NCTC 11168 reference genome [accession: AL11168.1; Supplementary Data 3][56,111,112]. A maximum likelihood phylogeny was constructed using RAxML v8.2.11[113] with the GTRGAMMA model (https://microreact.org/project/CampyIBS-CF).

Recombination regions inferred by ClonalFrame-ML[114] were masked from the core genome sequence alignment (i.e., replaced with gaps) using a custom script (https://github.com/kwongj/cfml-maskrc) to represent the clonal frame, against which associations were compared to correct for clonal descent and lineage effects[115]. Therefore, associations identified in multiple genetic backgrounds (lineages/STs) will be given greater importance and enriched in our analysis. Variants associated with PI-IBS among the three empirical association scores are deemed to be statistically significant, and candidates for biological association, pending subsequent confirmatory analyses. We prioritized IBS-associated variants that returned a subsequent p-value < 0.05 (which incorporates the phylogeny and corrected for multiple test using Bonferroni correction) or genes that contained over 100 associated variants (p > 0.05 in any of the three tests) for further investigation.

**Risk score**. A simple risk score was calculated based on the presence of our most significant pGWAS associations, similar to the score previously described to estimate risk of gastric cancer as a result of infection by different *Helicobacter pylori* genotypes[85]. Briefly, we screened 1000 variants with the most significant subsequent scores (Bonferroni corrected p-value ≤ 0.0075). By plotting the accuracy of each variant to predict the onset of PI-IBS in our dataset, we drew a threshold in upper tail of the distribution. This provided GWAS-informed markers for risk of developing PI-IBS. A cumulative score was then calculated by weighting the presence of each variant according to its subsequent score from treeWAS, which incorporates the direction of association, as scores with positive values representing an association for that variant with PI-IBS and a negative value representing an association with the control group.

**Epithelial cell culture and polarization**. T84 human colon cancer cells (from ATCC CCL-248) were cultured in DMEM/F12 medium (GIBCO, US) supplemented with 5% heat-inactivated fetal bovine serum (Life Technologies, US). No antibiotic supplementation was used. Cells were routinely passaged in 75 cm$^2$ flasks (Corning, US) at 5% $CO_2$, 37 °C. For the invasion assay, T84 cells were split into 96-well plates at a density of $10^5$ cells/well. For polarization, T84 cells were seeded at density of $7.5 \times 10^5$ cells/well into the apical chamber of 12-well Transwells (12 mm diameter; 3 μm pore size, Corning, US). The culture medium, DMEM/F12 medium supplemented with 5% heat inactivated FBS, was replaced every 2–3 days. Transepithelial electric resistance (TER) was measured manually every 2–3 days using an epithelial voltohmmeter 2 connected to a STX2 chopstick electrode (World Precision Instruments, USA) until the reading reached above 750 Ω cm$^2$. TER values for monolayers were calculated by subtracting the reading of an unseeded well from the seeded wells and by subsequent correction for surface area.

**Bacterial invasion**. Invasion assays (or gentamicin protection assay) were performed using established protocols[116,117]. Briefly, epithelial cells grown in 96-well plates, once reaching confluence, were infected with *C. jejuni* at concentration of $10^7$ CFU/well (i.e., multiplicity of infection of 100). The plates were centrifuged at $600 \times g$ for 5 min to promote contact of *C. jejuni* with host cells. After a 3-h incubation in a normal humidified incubator (5% $CO_2$, 37 °C), wells were washed five times with 1× phosphate-buffered saline to remove non-adherent bacteria, and then split in two halves. One half of the wells (3 wells) were lysed with 0.25% (wt/vol) sodium deoxycholate (Sigma) for 15 min, and serial 10-fold dilutions of the lysates were plated on MH agar plates to determine the total number of adherent and invaded bacteria. Antibiotic-free cell culture medium with gentamicin (200 μg/mL, Sigma), proven to be effective in killing extracellular *C. jejuni*[116,118], was then added to the other half of the wells (3 wells) and incubated for 3 h in a humidified incubator (37 °C and 5% $CO_2$). Cells were then washed five times with phosphate-buffered saline and lysed with 0.25% sodium deoxycholate (15 min; Sigma). Serial 10-fold dilutions of the lysates were plated on MH agar plates to determine the number of invaded bacteria. During each invasion experiment, highly invasive *C. jejuni* strain 81–176[117] and genome-sequenced low invasive strain NCTC 11168[119] were used as positive and negative controls, respectively. For each *C. jejuni* isolate, independent experiments were performed in triplicate and each triplicate on three different days, for a total of nine observations per strain.

**CellZScope system for *C. jejuni*-T84 interaction**. CellZscope is an automated, parallel, real-time recording system based on impedance spectroscopy that can be used for cells grown on porous filters[120]. T84 epithelial cell lines can be used as in vitro model systems of functional epithelial barriers. Polarized T84 monolayers grown on transwells (3 μm pore size, Corning, US) were transferred into a 24-well cell module of a CellZscope system (NanoAnalytics, Germany) with DMEM/F12 medium without serum. The 24-well cell module of CellZScope was placed in a humidified incubator (37 °C and 5% $CO_2$) overnight to allow equilibration. TER values were recorded every 40 min. The next day, *C. jejuni* cells were added to the apical chamber at the density of $2 \times 10^8$ CFU/well. This infection dose was used because a pilot experiment proved that lower than this dose did not result in significant TER change at 24 h post infection for most *C. jejuni* strains. At 24 h post infection, medium fractions from both apical and basolateral chambers were collected for cytokine analysis. The TER data was extracted, and TER change was

reflected by the percentage of TER at 24 h post incubation versus the initial. For each experiment, 3 replicates were measured.

**Cytokine analysis**. Apical and basolateral post-infection media fractions at the end of the 24 h CellZScope experiment were pelleted and bacteria removed with a 0.22 μm filter (Spin-X, Corning). Chemokines and cytokines in the media were measured using custom multiplex ELISAs (EMD Millipore, Darmstadt, Germany) for the following cytokines: IL-1β, IL-4, IL-6, IL-8, IL-10, TNFα, IFNγ, MCP-1, IP-10, MIG, and I-TAC. Samples were run in duplicate and average values were calculated and used for comparisons.

**Statistics and reproducibility**. All the results of genomics and GWAS analyses can be reproduced with the parameters indicated. In vitro studies of *C. jejuni* adhesion and invasion were performed in triplicate and each triplicate on three different days, for a total of nine observations per strain. For effects on barrier function in CellZScope, each strain was studied in triplicate.

Mean and standard deviation are reported for continuous variables, whereas frequencies and percentages are reported for categorical variables. A two-sided Mann–Whitney U-test assuming non-Gaussian distribution was used to compare variables between groups. Spearman's correlation (nonparametric) was used to analyze the strength of relationship between paired data. All analyses were done using GraphPad Prism 7 (GraphPad Software, San Diego, CA, USA). A p-value of <0.05 was considered significant.

**Reporting summary**. Further information on research design is available in the Nature Research Reporting Summary linked to this article.

## Data availability
Short read data are available on the NCBI Sequence Read Archive, associated with BioProject PRJNA675124 http://www.ncbi.nlm.nih.gov/bioproject/675124. Assembled genomes, GWAS summary files, and supplementary material are available from FigShare: https://doi.org/10.6084/m9.figshare.12493106. Raw data availability: Fig. 1b–e (Supplementary Data 4), Fig. 2d (Supplementary Data 6), Fig. 3b–e, Fig. 4 (Supplementary Data 9), Fig. 5f (Supplementary Data 10), Fig. 6 (Supplementary Data 9). Phylogenetic trees can be visualized and manipulated on Microreact for the whole dataset: https://microreact.org/project/CampyIBS and the recombination-free phylogeny used in the GWAS at https://microreact.org/project/CampyIBS-CF.

## Code availability
Scripts for manipulation of additional PIRATE outputs and treeWAS helper scripts are available on GitHub: https://github.com/SionBayliss.

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

## Acknowledgements

The study was funded by NIDDK K23 DK103911, R03 120745, and Department of Medicine K2R Program Award to M.G. Authors B.P., S.C.B. and S.K.S. are supported by the Medical Research Council (MRC) grant MR/L015080/1. C.T.P is supported by USDA CRIS Project 2030-42000-055-00D. Authors acknowledge Ms. Lori Anderson for administrative support.

## Author contributions

S.P.: acquisition, analysis, and interpretation of data; drafted manuscript. B.P.: acquisition, analysis, and interpretation of data; drafted manuscript. Z.W.: acquisition, analysis, and interpretation of data; revised manuscript. S.C.B.: acquisition, analysis, and interpretation of data; revised manuscript. X.Z.: acquisition and interpretation of data; revised manuscript. A.E.: acquisition and interpretation of data; revised manuscript. S.V.G.: interpretation of data; revised manuscript. S.J.: interpretation of data; revised manuscript. J.K.C.: interpretation of data; revised manuscript. E.M.: interpretation of data; revised manuscript. R.P.: designed work; revised manuscript. T.W.: acquisition and interpretation of data; revised manuscript. M.D.: acquisition and interpretation of data; revised manuscript. D.B.: designed work; acquisition of data; revised manuscript. K.S.: acquisition and interpretation of data; revised manuscript. C.T.P.: analysis, and interpretation of data; revised manuscript. G.F.: interpretation of data; revised manuscript. Q.Z.: designed work; acquisition, analysis, and interpretation of data; revised manuscript. S.K.S.: conceptualized and designed work; acquisition, analysis, and interpretation of data; drafted manuscript. M.G.: conceptualized and designed work; acquisition, analysis, and interpretation of data; drafted manuscript. All authors have approved the submitted version and are personally accountable for author's own contributions.

## Competing interests

The authors declare no competing interests.
