## [Peer Review File · Communications Biology]

Reviewers' comments:

Reviewer #1 (Remarks to the Author):

The overarching goal of this research manuscript is to identify *C. jejuni* genomic features that promote the development of the development of post-infectious irritable bowel syndrome (PI-IBS). Development of symptoms and post-infectious sequelae, as acknowledged by the author, is dependent of both the host and the pathogen. This study focuses on the pathogen only. *C. jejuni* is the number one cause of bacterial foodborne disease in the majority of high-income countries. Its incidence warrant studying the role of this pathogen in PI-IBS. Using whole-genome sequencing and analysis of clinical strains isolated from patients developing PI-IBS symptoms and comparing them to strains that did not, the authors identified a number of genes/alleles that may contribute to the development of PI-IBS.

The authors have performed a multitude of assays to demonstrate/validate their findings. These assays represent an enormous effort that needs to be acknowledged.

Are they novel and will they be of interest to others in the community and the wider field?

A similar approach has been used to link genomic factors to the development of post-infectious sequelae (including Guillain-Barre syndrome) but to the best of my knowledge, this manuscript the first one to describe the use of *C. jejuni* pangenome approach to the study of genetic factors involved in the development of PI-IBS.

This manuscript is hypothesis-generating, representing an initial effort that paves the way to the association of *C. jejuni* Phenotype/genomic factors to the development of PI-IBS. As acknowledged by the authors, these hypotheses need to be confirmed/solidified by additional studies.

The role of O-glycosylation is highlighted in this manuscript. It would be helpful if the authors could present an analysis and comparison of the locus composition and the type of sugar attached to the flagella.

Similarly, line 435-437, mention the role of several classes of LOS loci (presenting sialic acids) in invasion and translocation. It would be helpful to present the LOS class of the strains studied. Are the PI-IBS associated isolates belongs to the LOS class A, B or C? Is it statistically significant?

Same comment for the capsule: Based on the genetic composition of the genes of the CPS biosynthesis locus, do there is a trend/association of a particular CPS type in the isolates linked to the development of PI IBS?

In the introduction, the authors mention that "

"the calculation of a risk score for individual genotypes that, with further validation, could be a basis for medical interventions aimed at preventing PI-IBS following *Campylobacter* infection". Please explain in the discussion about what medical intervention could be used based on the hypothesis generated in this study? Also, please explain when the medical intervention would occur? During infection, post-infection? Also, please explain the feasibility of the approach: if the treatment needs to be provided to the patients to prevent invasion, how the isolation, sequencing/typing of the *C. jejuni* strain could be performed in time to allow such therapy?

The manuscript generates hypothesis that will hopefully be validated by additional studies performed by other researchers.

We would also be grateful if you could comment on the appropriateness and validity of any statistical analysis, as well the ability of a researcher to reproduce the work, given the level of detail provided.

The statistical analysis appears appropriate.

Additional comments:

Line 104 replace "microaerophilic" with "microaerobic"

Line 119 replace "NCTC11168" with "NCTC 11168"

Line 449 replace "glycosylation" with "protein glycosylation"

Reviewer #2 (Remarks to the Author):

The manuscript titled "Campylobacter jejuni genotypes associated with post-infection irritable bowel syndrome" determines and compares at molecular level different Campylobacter jejuni and Campylobacter coli strains isolated in post-infection IBS and non post-infection IBS. The results obtained demonstrate a group of genetic variations associated with PI-IBS, especially in genes involved in bacterial stress response and biosynthetic pathways. On the other hand, no differences were observed in genes associated with Campylobacter virulence. In addition, no specific C. jejuni lineage dominated in PI-IBS. The study also demonstrated that PI-IBS Campylobacter isolates were more adhesive, invasive and disruptive in-vitro. That being said, the manuscript is well organized and well investigated. Additionally, experiments have been done with an appropriate sample sizes and the authors have performed an appropriately and rigorously statistical analysis. To improve the quality of this paper, the authors should revise it according to the suggestions.

Reviewer #3 (Remarks to the Author):

Title. Campylobacter jejuni genotypes associated with post-infection irritable bowel syndrome

Summary. This manuscript investigates the genotypic factors that may be associated with post-infection irritable bowel syndrome (PI-IBS) using pangenome-wide association studies and phenotypic assays comparing C. jejuni isolates from patients who developed PI-IBS with those who did not. The authors identified variation in a number of genes that were potentially associated with PI-IBS. In vitro assays demonstrated greater adhesion, invasion, IL-8 and TNF alpha secretion on colonocytes with PI-IBS compared to PI-no-IBS strains. The authors highlight that specific Campylobacter genotypes confer greater in vitro virulence and increased risk of PI-IBS. These findings are useful for correlating Campylobacter genotypes with virulence and increased risk of PI-IBS.

Major Comments to the Author:-

This is an interesting study, combining genomics, GWAS and phenotypic assays to associate genotype to phenotype. I do have some major issues with the manuscript, in particular the results section (outlined below). Generally, I believe the results section can be improved in terms of accuracy, better description, and on occasions the claims are not justified. The results will need major improvements.

1. In terms of defining PI-IBS, I appreciate the authors are following specific criteria (Rome III), but what would be the significance of a patient developing PI-IBS in months 4-6 or 9-11? Does the specific range of 6-9 months potentially limit the complete understanding of PI-IBS?

2. Is there a distinction between typical gastroenteritis caused by PI-IBS? Given that gastroenteritis occurs within a few days after from contracting Campylobacter, here, PI-IBS is measured 6-9 months afterwards. What are the clinical manifestations of PI-IBS in comparison to the normal gastroenteritis? Given the focus of the manuscript is PI-IBS, I would have expected more description of this within the introduction.

3. Line 91 refers to the 'Table 1'. Is this meant to refer to Supplementary Table 1? The authors should

really check all of their designations (particularly the Supplementary material) as this is incredibly annoying when one is having to search and find the incorrect material. Line 123 - Is this meant to refer to Supplementary file 2? Does line 137 need to link to a Supplementary file? Is Line 158 supplementary file correct?

4. Either within the methods or the results, what is missing is a statement about how many patients were involved within the study, how many developed PI-IBS and how many were controls. Currently I am having to spend considerable time scouring the supplementary materials looking for this basic information. The authors should also describe in simple terms where these samples were isolated from e.g., hospital/location.

5. Line 93 states 'Data was missing for many patients'. In reference to what? How many were missing? Such sentences appear many times throughout the results sections and leave the reader scratching their head.

6. Line 95 states 'Patients with IBS prior to the infection or history of inflammatory bowel disease, celiac disease, and microscopic colitis were excluded to give a total of 49 C. jejuni isolates associated with the development of PI-IBS and 30 with controls (PI-no-IBS)'. However, this is somewhat confusing that Supplementary Table 1 has a total of 107 patients. I think the Supplementary Table 1 may need to highlight which samples were excluded?

7. Line 96 - 'The mean (SD) IBS symptom severity score (0-500 scale) for PI-IBS was 182.6 (117.6)'. Where is this information for each patient? How is this calculated and what is its significance?

8. Line 100 - 'C. jejuni isolates, collected via swabs and suspended in PBS'. Swabs were collected from where?

9. The sequencing section of the methods lacks depth. Users cannot reproduce this methodology if they wished. There is no mention of quality controls, contig ordering, annotation, genome visualisation?

10. Line 257 - The sentence 'Acute infection was predominantly caused by host generalist (n=27; ST21 and ST45 clonal complex 258 isolates) and poultry specialist (n=16; ST353, ST354, ST257) clonal complexes'. Where is this information in Figure 1? Or is this a general statement? A little confusing.

11. The selection of virulence genes in Figure 1A is unclear. In the methods, it states 'MLST analysis and examinations of known virulence associated genes and antibiotic resistance genes were carried out using pipeline SRST2, which takes Illumina reads as the input'. However, there is no information of how the genes in the Fig 1A were selected. Were they cherry picked? Why and how were they chosen? In addition, the authors have listed for example the presence/absence of the Type VI Secretion System. How was this screened for? The classical hcp gene or the full 13 gene loci? It is well documented for example that screening solely for the hcp in Campylobacter is inaccurate.

12. From Figure 1, there does not look like any significant findings? Control and PI-IBS strains come from many different types of clonal complexes. In addition, there does not seem to be any correlation with Generalist, Poultry, Ruminant or wild bird. Yes, the PI-IBD was higher in generalist. But what is the significance of this? Also, from the relatively low sample numbers, can we derive any conclusions?

13. I do not believe the heading for results section for Line 276 is justified; 'Pathogenic Campylobacter have a large pangenome'. Does the paragraph actually contain data that come to this conclusion? I do not believe the paragraph contains results that come to this conclusion. This happens a number of times throughout the results sections and claims are not always justified. Another example is 'Risk is higher following infection with specific lineages'. How is this statement

valid, given that in Figure 1, there were no associations between CC and control/PI-IBD? In Figure 3, the risk seems pretty high for the majority of CCs. In addition, the sample numbers for each CC are very low. Can the authors really make such overarching statements? It is also difficult to view each CC to each specific colour. Authors should improve this, or at least, re-order the legend to match the order of colour on the tree.

14. Figure S1E uses ROARY. There is nothing in the text that mentions this software. In fact, the authors have used a completely different tool for virulence gene detection. More information is needed within the methods as to why ROARY was used and specifically what it was used for. In addition, Figure S1E is not described well in the manuscript i.e., the two parts of the image.

15. Line 367 'Greater TER drop (% from baseline) was seen with PI-IBS strains vs. control strains (PI-IBS: $90.07\% \pm 7.98$ vs. control: $83.79\% \pm 16.05$, $p=0.058$) at 24 hours post-infection (Figure 4D)'. If this is correct, where is the significance?

Minor comments

1. PRJNA675124 not found. I hope it will be made available in the future.
2. Fig 1C label is missing.
3. Figure S1E is not described in the text.
4. Figure S1 legend needs more description.
5. Figure 4 panels ordered wrongly.

COMMUNICATIONS BIOLOGY MS #20-3348-T

Response to Reviewers' Comments

Reviewer #1

The overarching goal of this research manuscript is to identify C. jejuni genomic features that promote the development of the development of post-infectious irritable bowel syndrome (PI-IBS). Development of symptoms and post-infectious sequelae, as acknowledged by the author, is dependent of both the host and the pathogen. This study focuses on the pathogen only. C. jejuni is the number one cause of bacterial foodborne disease in the majority of high-income countries. Its incidence warrant studying the role of this pathogen in PI-IBS. Using whole-genome sequencing and analysis of clinical strains isolated from patients developing PI-IBS symptoms and comparing them to strains that did not, the authors identified a number of genes/alleles that may contribute to the development of PI-IBS.

The authors have performed a multitude of assays to demonstrate/validate their findings. These assays represent an enormous effort that needs to be acknowledged.

Are they novel and will they be of interest to others in the community and the wider field?

Response: We thank the reviewer for their review and positive comments. We strongly believe that the genomic and *in vitro* data are novel with several implications for advancing mechanistic understanding as well as care of patients with post-infection irritable bowel syndrome (PI-IBS). These findings will pave the way for *in vitro* and *in vivo* mechanistic studies to determine the role of these virulence elements in pathophysiology of PI-IBS. The GWAS-based risk-score upon validation and in combination with *in vitro* virulence characteristics will be useful in identifying infection cases that are at highest risk for developing *C. jejuni* PI-IBS. Considering there are no detailed studies examining the role of pathogenic virulence in PI-IBS pathophysiology, we believe these findings will be of significant interest to gastroenterology, microbiology and microbial genetic epidemiology fields. Furthermore, they have implications for improving food-safety that can ultimately abrogate the risk of acquiring *C. jejuni* enterocolitis.

A similar approach has been used to link genomic factors to the development of post-infectious sequelae (including Guillain-Barre syndrome) but to the best of my knowledge, this manuscript the first one to describe the use of C. jejuni pangenome approach to the study of genetic factors involved in the development of PI-IBS.

Response: The reviewer is correct. This is the first report using genomic/pangenomic approach to determine genes and variants associated with PI-IBS development. Furthermore, this manuscript is the first to present comprehensive *in vitro* virulence data (adhesion, invasion, barrier disruption, cytokine induction) on a large number of *C. jejuni* strains associated with PI-IBS or control status.

This manuscript is hypothesis-generating, representing an initial effort that paves the way to the association of C. jejuni Phenotype/genomic factors to the development of PI-IBS. As

acknowledged by the authors, these hypotheses need to be confirmed/solidified by additional studies. The role of O-glycosylation is highlighted in this manuscript. It would be helpful if the authors could present an analysis and comparison of the locus composition and the type of sugar attached to the flagella. Similarly, line 435-437, mention the role of several classes of LOS loci (presenting sialic acids) in invasion and translocation. It would be helpful to present the LOS class of the strains studied. Are the PI-IBS associated isolates belongs to the LOS class A, B or C? Is it statistically significant?

Same comment for the capsule: Based on the genetic composition of the genes of the CPS biosynthesis locus, do there is a trend/association of a particular CPS type in the isolates linked to the development of PI IBS?

Response: We thank the reviewer for these thoughtful comments, which have helped bring several aspects of the manuscript together. LOS and CPS type distribution have been added to **Figures 1A, D, E** (and **Table S3**). Additionally, they have been analyzed in the context of risk-score added as **Figures 3D, E**. Details of *in silico* determination have been included in the methods section (pages 6-7), loci classes summarised in the results section (pages 13-14) and discussed (pages 20-21). Determination of LOS and capsule types has been helpful in supporting our GWAS findings and helped explain the variation we see in specific genes related to some of the phenotypes tested in the laboratory.

We requested assistance from a collaborator for this additional work, who has now been added as a co-author (Dr. Craig T. Parker, USDA).

In the introduction, the authors mention that “the calculation of a risk score for individual genotypes that, with further validation, could be a basis for medical interventions aimed at preventing PI-IBS following Campylobacter infection”. Please explain in the discussion about what medical intervention could be used based on the hypothesis generated in this study? Also, please explain when the medical intervention would occur? During infection, post-infection? Also, please explain the feasibility of the approach: if the treatment needs to be provided to the patients to prevent invasion, how the isolation, sequencing/typing of the C. jejuni strain could be performed in time to allow such therapy?

Response: Thanks for raising this important point. Our recently published epidemiological study (*Characteristics and Risk Factors of Post-Infection Irritable Bowel Syndrome After Campylobacter Enteritis Clin Gastroenterol Hepatol* 2020 Jul 23) showed that time between onset of symptoms and start of antibiotic therapy may be important for subsequent risk of IBS. Patients who developed PI-IBS had a longer duration between onset of symptoms and start of antibiotic therapy. We also learnt that young age, females, history of hospitalization, bloody stools and abdominal cramps during the acute infection independently predicted PI-IBS development while fever was associated with a reduced risk. So, potentially patients who have these high-risk demographic features and infection with strains with high invasion and IL-8 production potential might need to be treated with antibiotics sooner. In another study, we have published that 40% of *C. jejuni* PI-IBS patients have significantly reduced microbial diversity and high luminal proteases (*Serine proteases as luminal mediators of intestinal barrier dysfunction and symptom severity in IBS Gut.* 2020 Jan;69(1):62-73). These patients can

potentially benefit from microbial replacement strategies either using probiotics or microbiota transplantation. It needs to be investigated in subsequent studies if certain host or pathogen characteristics may predispose to these downstream changes. Lastly, genomic sequencing is becoming more common. For example, the Minnesota Department of Health is now performing whole genome sequencing of as many samples as possible. In future, we will be able to validate our findings in larger cohorts which will make even a stronger case of post-infection interventions in population-based cohorts aimed at preventing establishment of long-term GI dysfunction. This has been discussed on page 24.

The manuscript generates hypothesis that will hopefully be validated by additional studies performed by other researchers. We would also be grateful if you could comment on the appropriateness and validity of any statistical analysis, as well the ability of a researcher to reproduce the work, given the level of detail provided. The statistical analysis appears appropriate.

Response: We have expanded on methodological details of the risk-score (page 8) as well as microbial sequencing and GWAS (pages 5-8). *In vitro* methods are already described in detail. We believe this should make it much easier for other researchers to use this approach in additional patient cohorts. The statistical analyses are appropriate and valid. Overall, we believe that readers should now have all the needed details to replicate and validate our results in independent cohorts.

Additional comments:

Line 104 replace “microaerophilic” with “microaerobic”

Response: This has been replaced.

Line 119 replace “NCTC11168” with “NCTC 11168”

Response: This has been replaced.

Line 449 replace “glycosylation” with “protein glycosylation”

Response: This part of text has been deleted.

Reviewer #2

The manuscript titled “Campylobacter jejuni genotypes associated with post-infection irritable bowel syndrome” determines and compares at molecular level different Campylobacter jejuni and Campylobacter coli strains isolated in post-infection IBS and non post-infection IBS. The results obtained demonstrate a group of genetic variations associated with PI-IBS, especially in genes involved in bacterial stress response and biosynthetic pathways. On the other hand, no differences were observed in genes associated with Campylobacter virulence. In addition, no specific C. jejuni lineage dominated in PI-IBS. The study also demonstrated that PI-IBS Campylobacter isolates were more adhesive, invasive and disruptive in-vitro. That being said, the manuscript is well organized and well investigated. Additionally, experiments have been done with an appropriate sample sizes and the authors have performed an appropriately and

rigorously statistical analysis. To improve the quality of this paper, the authors should revise it according to the suggestions.

Response: We thank the reviewer for their time in providing this thoughtful review and the encouraging comments.

Materials and Methods

Lane 91: Table 1 does not contain demographic and clinical symptom data.

Response: **Table 1** has been provided with these details.

Lane 92 to 97: these are results and should be presented in the results section.

Response: Text has been deleted since **Table 1** provides those results.

Lane 96: How the IBS symptom severity score was calculated?

Response: IBS-Symptom severity score (IBS-SSS) is a validated and widely used scoring for severity of IBS symptoms. It's a cumulative score of 5 questions (how severe is abdominal pain, number of days over last 10 days with pain*10, how severe is abdominal distension/tightness, how satisfied are you with bowel habits, how much IBS is affecting with life in general) each answered on a visual analogue scale from 0-100. The cumulative score of the 5 questions is then presented providing a total score range of 0-500. A cumulative score of 75-175 is categorized as mild IBS, 175-300 as moderate, and >300 as severe IBS. Text has been edited to indicate that average IBS-SSS of PI-IBS patients in this study fell in moderate severity category. Reference has been provided which lists the detailed questionnaire for interested readers (Francis CY, *Aliment Pharmacol Ther* 1997 Apr;11(2):395-402) (Page 12).

Lane 101: Please add the reference for C. jejuni sheep blood agar plates

Response: This has been provided (page 5).

Lane 123: There is no Supplementary file 1

Lane 126: There is no Supplementary file 2

Response: We apologize for our oversight in failing to upload these with the original submission. **Supplementary files 1-3** have been uploaded now.

Lane 169: This info should be part of the results, how the p-value cutoff was defined?

Response: Additional detail has been added to the methods describing the suite of statistical test that treeWAS performs and our rationale for prioritizing the subsequent p-values, while also considering hot spots of association (page 8).

Lane 174: Add ATCC reference number

Response: This has been added (page 8).

Results

Lane 259: ST61 clonal complexes is not described in Figure S1A or B

Response: These have been described in **Supplementary Figure 1D**.

Lane 286: Figure S1E should have a better resolution, I suggest a separate file only for this Figure.

Response: As suggested by the reviewer, this has been separated into **Figure S2**.

Lanes 289 to 292: Please identify the clusters in the Figure, it is hard to see it.

Response: These have been highlighted in the updated stand-alone **Figure S2**.

Lane 296: The gene tetO is not described in figure 1A.

Lane 298: the resistance elements aph(3')-III and sat4A are not in Figure 1A.

Response: These have been added to **Figure 1A**.

Lane 301: Are peb1 and jlp1 same as nbA and jLpA?

Response: peb1 is same as nbA. jLp1 has been removed from updated figure and the text.

Lane 319: Is the p-value cutoff correct? Why 0.05 is accepted as significant? How the cutoff was defined?

Response: It is actually “Bonferroni corrected p-value” and text has been edited to reflect that (page 15). More detail added to the methods on how treeWAS calculates statistical significance (page 8). All p-values below 0.05 are reported in the **Supplementary Table 4**, but only our most significant (and /or genes that contain the most associated variants) were investigated further.

Lane 322: The is no Supplementary file S3.

Response: This has now been uploaded

Lane 344: How the p-value cutoff was defined?

Response: More detail added to the methods (page 8). Statistical p-values reported for all variants, but only the top 1,000 variants were carried over into this analysis (subsequent p-value \leq 0.0075) (page 16).

Lanes 368 and 369: TER drop is not significant.

Response: That is correct considering the p-value of 0.06. The text has been edited to reflect that. Additionally, subheading of the section has been edited removing the barrier part (page 17).

General question:

Which isolates specifically were used in the in-vitro tests? Which lineage?

Response: A random selected set of PI-IBS and control isolates were used for *in vitro* studies. These belonged to various lineages. The sequence types for these strains have been added to **Supplementary Table 6** so readers have easy access to this information.

Discussion

Lane 439: Had the isolates from ST22 clonal complex more potential for invasion in-vitro?

Response: Three of the isolates were ST22 clonal complex (all belonging to the PI-IBS cohort). The average adhesion, invasion and barrier drop with these 3 strains were 2.53E-03, 3.30E-03 and -93%. This was not different from group means. However, one of these 3 strains was more virulent (isolate ID 13 vs 1 and 11).

Lanes 498 and 499: this sentence is very confusing; I suggest re-phrase it.

Response: The sentence has been rephrased (page 21).

Lanes 510 and 511: the variants associated with disruption of epithelial barrier were not significant.

Response: Variants in Cj0139, a putative endonuclease significantly associated with barrier disruption. Hence, this statement is correct.

Lane 526: Is there a possibility that C. jejuni is in the small intestine?

Response: *C. jejuni* infection typically involves distal small bowel and colon. Our intent with this statement is that we do not see chronic colonization of *C. jejuni* in the gastrointestinal tract in PI-IBS patients. Statement has been edited to provide clarity (page 24).

Reviewer #3

Title. Campylobacter jejuni genotypes associated with post-infection irritable bowel syndrome

Summary. This manuscript investigates the genotypic factors that may be associated with post-infection irritable bowel syndrome (PI-IBS) using pangenome-wide association studies and phenotypic assays comparing C. jejuni isolates from patients who developed PI-IBS with those who did not. The authors identified variation in a number of genes that were potentially associated with PI-IBS. In vitro assays demonstrated greater adhesion, invasion, IL-8 and TNF alpha secretion on colonocytes with PI-IBS compared to PI-no-IBS strains. The authors highlight that specific Campylobacter genotypes confer greater in vitro virulence and increased

risk of PI-IBS. These findings are useful for correlating Campylobacter genotypes with virulence and increased risk of PI-IBS.

Major Comments to the Author:

This is an interesting study, combining genomics, GWAS and phenotypic assays to associate genotype to phenotype. I do have some major issues with the manuscript, in particular the results section (outlined below). Generally, I believe the results section can be improved in terms of accuracy, better description, and on occasions the claims are not justified. The results will need major improvements.

Response: Thank you for your positive review and raising these important points. We have made significant changes to the manuscript and especially the result section clarifying headings, providing better description in the text and figures as well as more accurately claiming what the data is showing. Details as noted in response to the specific questions below.

In terms of defining PI-IBS, I appreciate the authors are following specific criteria (Rome III), but what would be the significance of a patient developing PI-IBS in months 4-6 or 9-11? Does the specific range of 6-9 months potentially limit the complete understanding of PI-IBS?

Response: Diagnosis of IBS is based on chronicity of symptoms. For example if an individual has symptoms for a month, we cannot define them as having IBS as there hasn't been sufficient chronicity. For this reason, the Rome questionnaire asks for "onset of symptoms 6 months ago or more". So, ongoing symptoms for 9-11 months will be captured in the Rome criteria. If someone has symptoms for 4 months and they resolve completely, then they won't be categorized as IBS. This is the reason why we survey individuals at 6-9 months following the infection. Rome criteria are rigorously validated and internationally accepted as best measures for categorizing IBS. Hence, the use of these criteria is a strength of the current study. This has been explained in more detail on page 12.

Is there a distinction between typical gastroenteritis caused by PI-IBS? Given that gastroenteritis occurs within a few days after from contracting Campylobacter, here, PI-IBS is measured 6-9 months afterwards. What are the clinical manifestations of PI-IBS in comparison to the normal gastroenteritis? Given the focus of the manuscript is PI-IBS, I would have expected more description of this within the introduction.

Response: PI-IBS is a delayed sequelae of *C. jejuni* infection where patients suffer from abdominal pain or discomfort at least 2-3 times a month and have either altered frequency or form of bowel movements at least 25% of the time. This is different from acute *C. jejuni* gastroenteritis which is an acute episode lasting 3-10 days where patients have significant abdominal pain and considerable diarrhea. They often have fever and some may need to be hospitalized. However, most patients recover completely from *C. jejuni* enteritis whereas a subset develops PI-IBS. The details describing PI-IBS have been added to the introduction (page 3). Additionally, the added **Table 1** provides spectrum of symptoms associated with *C. jejuni* enterocolitis.

Line 91 refers to the 'Table 1'. Is this meant to refer to Supplementary Table 1? The authors should really check all of their designations (particularly the Supplementary material) as this is incredibly annoying when one is having to search and find the incorrect material. Line 123 - Is this meant to refer to Supplementary file 2? Does line 137 need to link to a Supplementary file? Is Line 158 supplementary file correct?

Response: That section has been moved to Results (page 12) and linked to the appropriate **Table 1**. We regret for the oversight and all main and Supplementary Tables have now been accurately uploaded and linked. **Supplementary files 1-3** have also been uploaded and available for your review.

Either within the methods or the results, what is missing is a statement about how many patients were involved within the study, how many developed PI-IBS and how many were controls. Currently I am having to spend considerable time scouring the supplementary materials looking for this basic information. The authors should also describe in simple terms where these samples were isolated from e.g., hospital/location.

Response: Page 12 and **Table 1** describes the number of patients and their PI-IBS (n=49)/control (n=30) status. These patients deposited the stool sample either to a clinic or a hospital within the state of Minnesota which led to the diagnosis of *Campylobacter* enterocolitis. Subsequently, these samples were shipped to the Minnesota Department of Health as a CDC requirement for the state. We obtained the samples from central laboratory of the Minnesota Department of Health. These details have been added to Page 12 (Results; 1st section).

Line 93 states 'Data was missing for many patients'. In reference to what? How many were missing? Such sentences appear many times throughout the results sections and leave the reader scratching their head.

Response: That vague statement has been removed. The **Table 1** for clinical symptoms and the footnote in that Table mentions exactly the number of patients with missing data for each symptom. This data is derived from an interview conducted by the Minnesota Department of Health shortly after the acute infection was reported. As with interviews and surveys, several patients choose to not answer a particular question resulting in a missing response. Also, this section has been moved to Results (page 12).

Line 95 states 'Patients with IBS prior to the infection or history of inflammatory bowel disease, celiac disease, and microscopic colitis were excluded to give a total of 49 C. jejuni isolates associated with the development of PI-IBS and 30 with controls (PI-no-IBS)'. However, this is somewhat confusing that Supplementary Table 1 has a total of 107 patients. I think the Supplementary Table 1 may need to highlight which samples were excluded?

Response: **Supplementary Table 1** has 94 patients (not 107). 15 samples were excluded (6 *C. coli*; 9 PI-IBS or control data was not available). This leaves 79 patients of which 49 were PI-IBS associated and 30 controls. A *footnote* has been added to **Supplementary Table 1** to clarify this information. Additionally, a column for *Campylobacter* species has been added to the

Supplementary Table 1 so it is easy for the reader to follow. Further, the text on page 12 has been edited to clarify.

Line 96 - 'The mean (SD) IBS symptom severity score (0-500 scale) for PI-IBS was 182.6 (117.6)'. Where is this information for each patient? How is this calculated and what is its significance?

Response: IBS-Symptom severity score (IBS-SSS) is a validated and widely used scoring for severity of IBS symptoms. It's a cumulative score of 5 questions (how severe is abdominal pain, number of days over last 10 days with pain*10, how severe is abdominal distension/tightness, how satisfied are you with bowel habits, how much IBS is affecting with life in general) each answered on a visual analogue scale from 0-100. The cumulative score of the 5 questions is then presented providing a total score range of 0-500. A cumulative score of 75-175 is categorized as mild IBS, 175-300 as moderate, and >300 as severe IBS. Edit has been made to indicate that average IBS-SSS of PI-IBS patients in this study fell in moderate severity category. Reference has been provided which lists the detailed questionnaire for interested readers (Francis CY, *Aliment Pharmacol Ther* 1997 Apr;11(2):395-402) (Page 12).

Line 100 - 'C. jejuni isolates, collected via swabs and suspended in PBS'. Swabs were collected from where?

Response: This statement has been removed as the method was used for PFGE which is not part of the manuscript.

The sequencing section of the methods lacks depth. Users cannot reproduce this methodology if they wished. There is no mention of quality controls, contig ordering, annotation, genome visualisation?

Response: Additional detail has been added throughout the methods section, including further clarification of sequencing methods (page 5-6), assembly, QC of assembled genomes, annotation and various visualization techniques (pages 6-7).

Line 257 - The sentence 'Acute infection was predominantly caused by host generalist (n=27; ST21 and ST45 clonal complex isolates) and poultry specialist (n=16; ST353, ST354, ST257) clonal complexes'. Where is this information in Figure 1? Or is this a general statement? A little confusing.

Response: **Figure 1B** has been edited to highlight the clonal complexes using different colors. Accordingly the text has been edited in results section to increase clarity (page 12). **Supplementary Fig 1D** also highlights these categories of isolates.

The selection of virulence genes in Figure 1A is unclear. In the methods, it states 'MLST analysis and examinations of known virulence associated genes and antibiotic resistance genes were carried out using pipeline SRST2, which takes Illumina reads as the input'. However, there is no information of how the genes in the Fig 1A were selected. Were they cherry picked? Why and how were they chosen? In addition, the authors have listed for example the

presence/absence of the Type VI Secretion System. How was this screened for? The classical hcp gene or the full 13 gene loci? It is well documented for example that screening solely for the hcp in Campylobacter is inaccurate.

Response: These genes were selected due to their known significance in *C. jejuni* virulence. All virulence genes identified in the dataset by ARG-ANOTT are included (minus LOS, capsule and motility genes, which were summarized by inclusion of LOS classes and Penner types). Further to the reviewers comment regarding the T6SS, we have also included details of the presence of two well characterized *Campylobacter* plasmids (pTet and pVir), one of which (pVir) contains the mobile T6SS. The presence of these plasmids is indicated in **Figure 1A** when 75% or more of the genes contained within these plasmids were identified.

From Figure 1, there does not look like any significant findings? Control and PI-IBS strains come from many different types of clonal complexes. In addition, there does not seem to be any correlation with Generalist, Poultry, Ruminant or wild bird. Yes, the PI-IBD was higher in generalist. But what is the significance of this? Also, from the relatively low sample numbers, can we derive any conclusions?

Response: We have incorporated several reviewer comments into an updated **Figure 1** and tried to emphasize our findings from molecular characterization of our dataset. Specifically, that [1] PI-IBS emerges from different genetic backgrounds, i.e. there is no single clone, plasmid or gene that is responsible for PI-IBS; and [2] that different lineages, either defined by sequence type (in this case clonal complex), LOS class or Penner serotypes, differ in their ability to initiate PI-IBS and have different risk profiles. This builds our rationale for embarking on a pGWAS analysis and also represents the most comprehensive characterization of any PI-IBS associated bacterial genomics collection to date.

I do not believe the heading for results section for Line 276 is justified; 'Pathogenic Campylobacter have a large pangenome'. Does the paragraph actually contain data that come to this conclusion? I do not believe the paragraph contains results that come to this conclusion. This happens a number of times throughout the results sections and claims are not always justified.

Response: This has been changed to “*Pathogenic Campylobacter have an open pan-genome*”

Another example is 'Risk is higher following infection with specific lineages'. How is this statement valid, given that in Figure 1, there were no associations between CC and control/PI-IBD? In Figure 3, the risk seems pretty high for the majority of CCs. In addition, the sample numbers for each CC are very low. Can the authors really make such overarching statements? It is also difficult to view each CC to each specific colour. Authors should improve this, or at least, re-order the legend to match the order of colour on the tree.

Response: We realize that our headings were not clear enough and these have now been edited. Our intention was not to say that risk is higher with different lineages. In fact we wanted to say that risk-score has variable predictive value for different strain lineages. The section title has been revised to reflect that. Edits made on pages 14, 16, 17 and 18.

Figure S1E uses ROARY. There is nothing in the text that mentions this software. In fact, the authors have used a completely different tool for virulence gene detection. More information is needed within the methods as to why ROARY was used and specifically what it was used for. In addition, Figure S1E is not described well in the manuscript i.e., the two parts of the image.

Response: Figure S1E [original submission] was constructed using PIRATE (not ROARY) and visualized in phandango (details added to page 6). PIRATE (and ROARY) estimate the size and composition of the core and accessory genomes. Both use annotated genomes as input, for which we used PROKKA. Virulence genes and AMR determinants are identified through BLAST searches of curated databases (VfDB, ARG-ANNOT). The results of these analyses can be visualized in microreact and/or phandango. These details have been further clarified in the methods. This figure can now be found as a standalone **Supplementary Figure S2** [new].

Line 367 'Greater TER drop (% from baseline) was seen with PI-IBS strains vs. control strains (PI-IBS: 90.07% ± 7.98 vs. control: 83.79% ± 16.05, p=0.058) at 24 hours post-infection (Figure 4D)'. If this is correct, where is the significance?

Response: This statement has been edited as with p-value of 0.06, threshold of significance has not been met. Additionally, section title has been revised (page 17).

Minor comments

PRJNA675124 not found. I hope it will be made available in the future.

Response: PRJNA675124 is available. Please use the link:
<https://www.ncbi.nlm.nih.gov/bioproject/?term=PRJNA675124>

Fig 1C label is missing.

Response: Figure 1C is labelled on x and y-axis. Additionally, the legend is available.

Figure S1E is not described in the text.

Response: This figure has now been separated into **Supplementary Figure 2 (Figure S2)** and its details have been provided in the text (page 14).

Figure S1 legend needs more description.

Response: The legends have been expanded with additional details added. Additionally S1E is now S2 with its own legend.

Figure 4 panels ordered wrongly.

Response: The panels have been corrected.

REVIEWERS' COMMENTS:

Reviewer #1 (Remarks to the Author):

Thanks for your revisions

Reviewer #3 (Remarks to the Author):

The authors have made considerable effort to improve the manuscript and I believe the manuscript is now much clearer and suitable for publication. Thank you to the authors for their efforts.